# KHNYN is essential for the zinc finger antiviral protein (ZAP) to restrict HIV-1 containing clustered CpG dinucleotides

Mattia Ficarelli[1], Harry Wilson[1†], Rui Pedro Galão[1†], Michela Mazzon[2], Irati Antzin-Anduetza[1], Mark Marsh[2], Stuart JD Neil[1*], Chad M Swanson[1*]

[1]Department of Infectious Diseases, School of Immunology and Microbial Sciences, King's College London, London, United Kingdom; [2]MRC Laboratory for Molecular Cell Biology, University College London, London, United Kingdom

**Abstract** CpG dinucleotides are suppressed in most vertebrate RNA viruses, including HIV-1, and introducing CpGs into RNA virus genomes inhibits their replication. The zinc finger antiviral protein (ZAP) binds regions of viral RNA containing CpGs and targets them for degradation. ZAP does not have enzymatic activity and recruits other cellular proteins to inhibit viral replication. We found that KHNYN, a protein with no previously known function, interacts with ZAP. KHNYN overexpression selectively inhibits HIV-1 containing clustered CpG dinucleotides and this requires ZAP and its cofactor TRIM25. KHNYN requires both its KH-like domain and NYN endonuclease domain for antiviral activity. Crucially, depletion of KHNYN eliminated the deleterious effect of CpG dinucleotides on HIV-1 RNA abundance and infectious virus production and also enhanced the production of murine leukemia virus. Overall, we have identified KHNYN as a novel cofactor for ZAP to target CpG-containing retroviral RNA for degradation.

DOI: https://doi.org/10.7554/eLife.46767.001

*For correspondence:
stuart.neil@kcl.ac.uk (SJDN);
chad.swanson@kcl.ac.uk (CMS)

†These authors contributed equally to this work

Competing interests: The authors declare that no competing interests exist.

## Introduction

A major component of the innate immune system are cell intrinsic antiviral proteins. These act at multiple steps in viral replication cycles and some are induced by type I interferons (*Schneider et al., 2014*). Many viruses have evolved mechanisms to evade inhibition by these proteins. First, viruses can encode proteins that counteract specific antiviral factors. Examples of this mechanism in HIV-1 are the accessory proteins Vif and Vpu that counteract APOBEC3 cytosine deaminases and Tetherin, respectively (*Malim and Bieniasz, 2012*). Second, viral protein or nucleic acid sequences can evolve to prevent recognition by antiviral factors. The abundance of CpG dinucleotides is suppressed in many vertebrate RNA virus genomes and when CpGs are experimentally introduced into picornaviruses or influenza A virus, replication is inhibited (*Atkinson et al., 2014*; *Burns et al., 2009*; *Gaunt et al., 2016*; *Karlin et al., 1994*; *Tulloch et al., 2014*). This shows that CpG suppression in diverse RNA viruses is required for efficient replication.

CpG dinucleotides are also suppressed in the HIV-1 genome and multiple studies have shown that they are deleterious for replication (*Antzin-Anduetza et al., 2017*; *Kypr et al., 1989*; *Shpaer and Mullins, 1990*; *Takata et al., 2017*; *Theys et al., 2018*; *Wasson et al., 2017*). Recently, the cellular antiviral protein ZAP was shown to bind regions of HIV-1 RNA with high CpG abundance and target them for degradation, which at least partly explains why this dinucleotide inhibits viral replication (*Takata et al., 2017*). ZAP (also known as ZC3HAV1) is a component of the innate immune response targeting viral RNAs in the cytoplasm to prevent viral protein synthesis (*Li et al., 2015*). ZAP inhibits the replication of a diverse range of viruses including retroviruses, alphaviruses, filoviruses, hepatitis B virus and Japanese encephalitis virus as well as retroelements (*Bick et al.,*

**eLife digest** Like many viruses, the genetic information of the human immunodeficiency virus (or HIV for short) is formed of molecules of RNA, which are sequences of building blocks called nucleotides. Once the virus is inside human cells, a protein called ZAP can identify viral RNAs by binding to a precise motif, a combination of two nucleotides called CpG. This allows the cell to destroy the viral RNA, thus preventing the virus from multiplying. However, HIV and other viruses that infect mammals are often able to 'hide' from ZAP because their genetic codes have many fewer CpG nucleotides than what would be expected by chance.

ZAP by itself does not appear to be able to cut up RNA, so it is thought that it recruits other, as yet unidentified, proteins to destroy the genome of viruses. Here, Ficarelli et al. used genetic techniques to identify a new human protein called KHNYN that interacts with ZAP.

First, a new version of the RNA genome of HIV was engineered, which contained higher numbers of CpGs: this CpG-enriched virus could be inhibited by ZAP in human cells. The experiments showed that increasing the amount of KHNYN protein led to lower levels of HIV genomes enriched in CpG. However, increasing the levels of KHNYN protein in mutant cells without ZAP had no effect on how well CpG-enriched HIV multiplied. CpG-enriched HIV and another related virus with many CpG nucleotides were able to multiply more successfully in mutant cells lacking the KHNYN protein than in normal cells. Further experiments also suggested that mutating a region of KHNYN which is likely to cut RNA prevented it from inhibiting HIV enriched with CpGs.

Artificially manipulating the CpG nucleotide content of viral sequences could help create viruses useful for human health. For instance, weakened viruses could be designed for use in vaccines. Some human tumors have decreased levels of ZAP, and it could therefore be possible to build viruses that healthy cells can destroy, but which could multiply in and kill cancer cells. However, before these approaches can be developed, exactly how ZAP and KHNYN degrade strands of viral RNA needs to be characterized.

DOI: https://doi.org/10.7554/eLife.46767.002

*2003*; *Chiu et al., 2018*; *Gao et al., 2002*; *Goodier et al., 2015*; *Mao et al., 2013*; *Moldovan and Moran, 2015*; *Müller et al., 2007*; *Takata et al., 2017*; *Zhu et al., 2011*). There are two human ZAP isoforms, ZAP-L and ZAP-S (*Kerns et al., 2008*). Both isoforms contain a N-terminal RNA binding domain containing four CCCH-type zinc finger motifs but ZAP-L also contains a catalytically inactive C-terminal poly(ADP-ribose) polymerase (PARP)-like domain (*Chen et al., 2012*; *Guo et al., 2004*; *Kerns et al., 2008*). Importantly, neither isoform of ZAP has nuclease activity and it likely recruits other cellular proteins to degrade viral RNAs. Identifying and characterizing these cofactors for ZAP is essential to understand how it restricts viral replication.

ZAP requires the E3 ubiquitin ligase TRIM25 for its antiviral activity against Sindbis virus and HIV-1 with clustered CpGs (*Li et al., 2017*; *Takata et al., 2017*; *Zheng et al., 2017*). While ZAP has been reported to interact with several components of the 5′−3′ and 3′−5′ RNA degradation pathways, depletion of these proteins did not substantially increase infectious virus production for HIV-1 containing clustered CpG dinucleotides (*Goodier et al., 2015*; *Guo et al., 2007*; *Takata et al., 2017*; *Zhu et al., 2011*). This suggests that additional proteins may be required for ZAP to inhibit viral replication. Herein, we identify KHNYN as a cytoplasmic protein that interacts with ZAP and is necessary for CpG dinucleotides to inhibit HIV-1 RNA and protein abundance.

## Results

### KHNYN interacts with ZAP and selectively inhibits HIV-1 containing clustered CpG dinucleotides in a ZAP- and TRIM25-dependent manner

To identify candidate interaction partners for ZAP, a yeast two-hybrid screen was performed for full-length ZAP-S and ZAP-L using prey fragments from a mixed Pam3CSK4-induced and IFNβ-induced human macrophage cDNA library. Candidate interacting proteins were assigned a Predicted Biological Score (PBS) of A to F (*Formstecher et al., 2005*): A = very high confidence in the interaction, B = high confidence in the interaction and C = good confidence in the interaction. Scores of D to F

are low confidence interactions, non-specific interactions or proven technical artifacts. For ZAP-S, 11 clones were obtained from 60.4 million tested interactions. 10 of these contained a prey fragment encoding KHNYN (*Figure 1*) and had a PBS = A. One clone had an insert encoding MARK3 but this was in the antisense orientation and therefore did not receive a score. For ZAP-L, two positive clones were analyzed from 104 million tested interactions. Both of these had an insert encoding KHNYN and had a PBS = C. KHNYN has two isoforms (KHNYN-1 and KHNYN-2) that contain a N-terminal KH-like domain and a C-terminal NYN endoribonuclease domain (*Figure 1*) (*Anantharaman and Aravind, 2006*). The selected interaction domain, which is the amino acid sequence shared by all prey fragments matching KHNYN, comprised amino acids 572–719 of KHNYN-2 for the clones identified in both screens. A yeast two-hybrid screen was then performed using the same library with full length KHNYN-2 as the bait. Nine clones were isolated that encode ZAP and these had a PBS = A. The selected interaction domain was amino acids 4–352, which is present in both isoforms (*Figure 1*). Supporting the reproducibility of this interaction, KHNYN has also been identified as a ZAP-interacting factor in large-scale affinity purification–mass spectrometry and in vivo proximity-dependent biotinylation (BioID) screens (*Huttlin et al., 2017*; *Youn et al., 2018*).

We first confirmed the interaction between ZAP and KHNYN by co-immunoprecipitation and found both KHNYN isoforms interacted with both isoforms of ZAP (*Figure 2A and B*). This interaction was RNase insensitive (*Figure 2C*). Since ZAP mediates degradation of HIV-1 RNAs with clustered CpG dinucleotides in the cytoplasm (*Takata et al., 2017*), its cofactors are likely to be localized in this compartment. Therefore, we analyzed the subcellular localization of KHNYN and observed that it localizes to the cytoplasm similar to ZAP (*Figure 2D*). Its localization was not affected when ZAP was knocked out using CRISPR-Cas9-mediated genome editing.

The mechanisms that allow a virus to escape the innate immune response often have to be inactivated to study the effect of antiviral proteins. For example, HIV-1 Vpu or Vif have to be mutated to allow Tetherin or APOBEC3 antiviral activity to be analyzed (*Malim and Bieniasz, 2012*). Since CpG dinucleotides are suppressed in HIV-1, endogenous ZAP does not target the wild-type virus (*Takata et al., 2017*). However, a ZAP-sensitive HIV-1 can be created by introducing CpGs through synonymous mutations into the *env* open-reading frame in the viral genome. This makes HIV-1 an excellent system to study the mechanism of action of this antiviral protein because isogenic viruses can be analyzed that differ only in their CpG abundance and therefore ZAP-sensitivity (*Takata et al., 2017*). To determine if KHNYN overexpression inhibited wild-type HIV-1 or HIV-1 with 36 CpG dinucleotides introduced into *env* nucleotides 86–561 (HIV-1$_{EnvCpG86-561}$) (*Figure 2—figure supplement 1*), each isoform was overexpressed in the context of a single cycle replication assay. As expected, transfection of the HIV-1$_{EnvCpG86-561}$ provirus into HeLa cells yielded substantially less infectious virus than wild-type HIV-1, which was accounted for by reduced expression of Gag and Env proteins (*Figure 2E and F*). While KHNYN-1 or KHNYN-2 overexpression decreased wild-type HIV-1 infectivity by ~5 fold, they decreased HIV-1$_{EnvCpG86-561}$ infectivity by ~400 fold (*Figure 2E*). The inhibition of

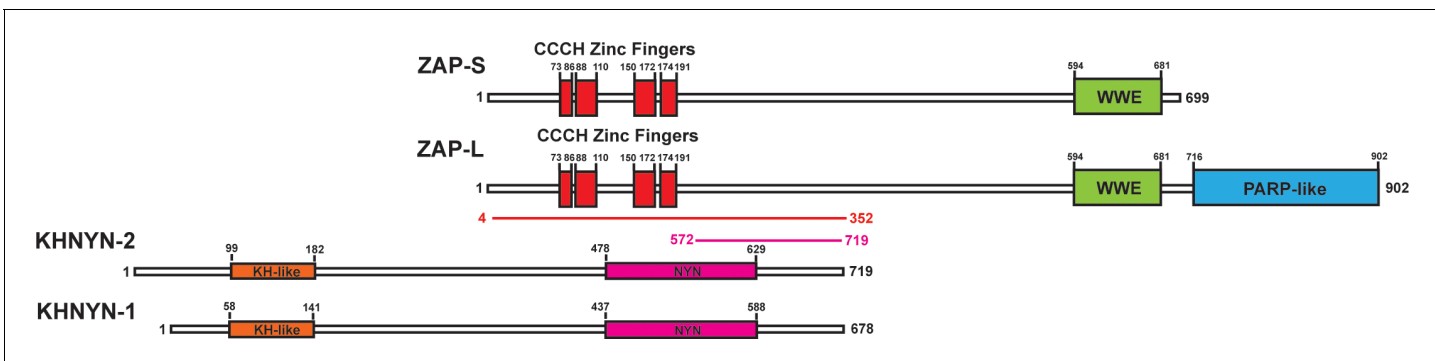

**Figure 1.** KHNYN is a ZAP-interacting factor identified by yeast two-hybrid screening. A yeast two-hybrid screen for ZAP-S and ZAP-L interacting factors identified a region in KHNYN-1 and KHNYN-2. The selected interaction domain (SID) is the amino acid sequence shared by all prey fragments and is shown in magenta. A reciprocal yeast two-hybrid screen using KHNYN-2 as the bait identified a region in ZAP-S and ZAP-L. The SID is shown in red.

DOI: https://doi.org/10.7554/eLife.46767.003

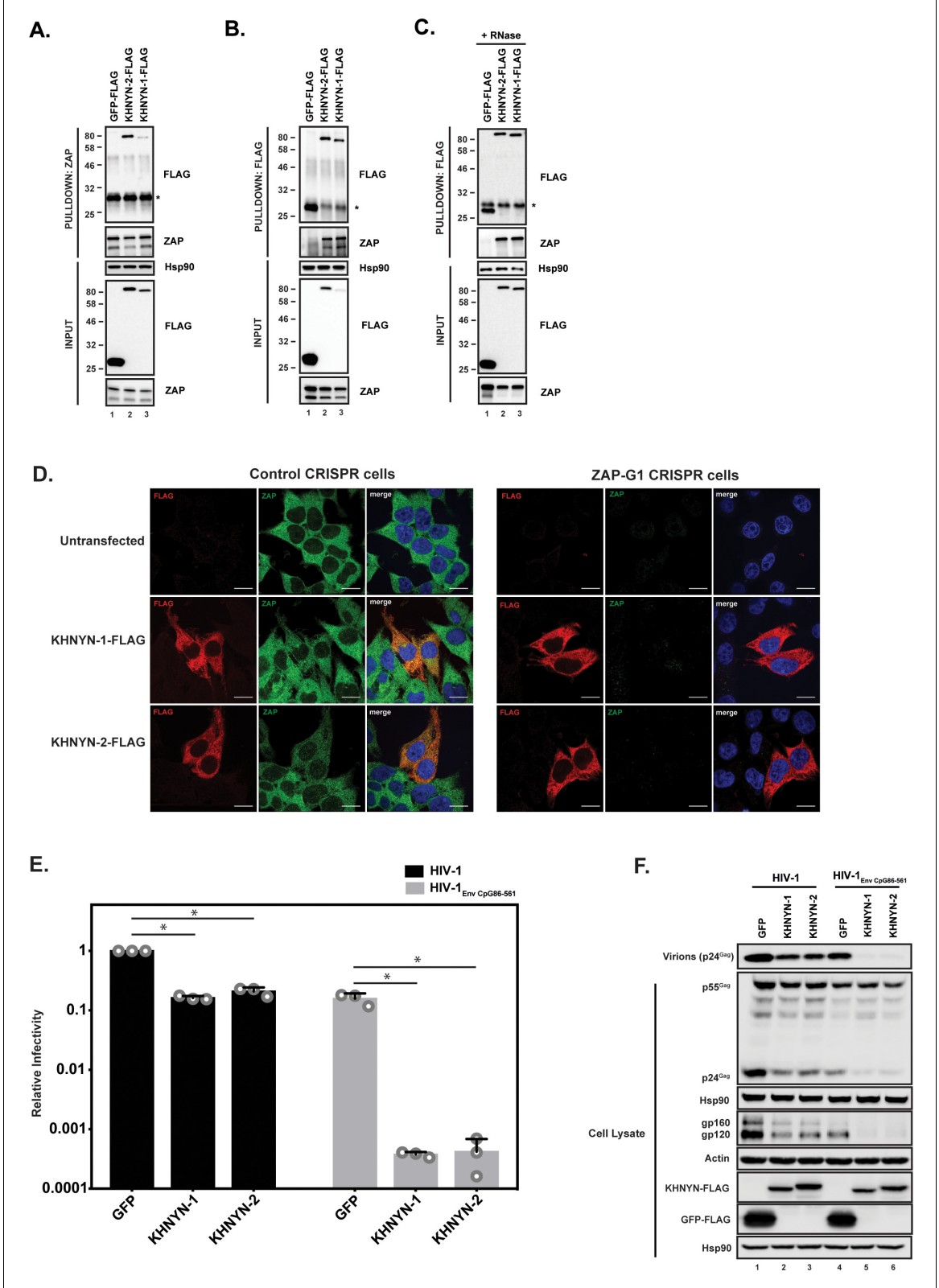

**Figure 2.** KHNYN interacts with ZAP and selectively inhibits HIV-1 containing clustered CpG dinucleotides. (**A**) Lysates of HEK293T cells transfected either with pGFP-FLAG, pKHNYN-1-FLAG or pKHNYN-2-FLAG were immunoprecipitated with an anti-ZAP antibody. Post-nuclear supernatants and immunoprecipitation samples were analyzed by immunoblotting for Hsp90, KHNYN-FLAG and ZAP. * indicates a non-specific band. (**B**) Lysates of HEK293T cells transfected either with pGFP-FLAG, pKHNYN-1-FLAG or pKHNYN-2-FLAG were immunoprecipitated with an anti-FLAG antibody. Post-

*Figure 2 continued on next page*

*Figure 2 continued*

nuclear supernatants and immunoprecipitation samples were analyzed by immunoblotting for HSP90, KHNYN-FLAG and ZAP. * indicates a non-specific band. (C) Lysates of HEK293T cells transfected with pZAP-L and either pGFP-FLAG, pKHNYN-1-FLAG or pKHNYN-2-FLAG were treated with RNase and then immunoprecipitated with an anti-FLAG antibody. Post-nuclear supernatants and immunoprecipitation samples were analyzed by immunoblotting for HSP90, KHNYN-FLAG and ZAP. * indicates a non-specific band. (D) Panels show representative fields for the localization of KHNYN-1-FLAG or KHNYN-2-FLAG and endogenous ZAP in either 293T Control CRISPR cells expressing a guide RNA targeting *LacZ* or 293T *ZAP* guide 1 (ZAP-G1) CRISPR cells. Cells were stained with an anti-FLAG antibody (red), anti-ZAP antibody (green) and DAPI (blue). The scale bar represents 10 μM. (E–F) HeLa cells were transfected with 500 ng pHIV-1 or pHIV-1$_{EnvCpG86-561}$ and 500 ng of pGFP-FLAG, pKHNYN-1-FLAG or pKHNYN-2-FLAG. See also *Figure 2—figure supplement 1*. Culture supernatants were used to infect TZM-bl reporter cells to measure infectivity (E). The bar charts show the average values of three independent experiments normalized to the value obtained for HeLa cells co-transfected with pHIV-1 and pGFP-FLAG. Data are represented as mean ± SD. *p<0.05 as determined by a two-tailed unpaired t-test. p-values for GFP verses KHNYN-1 and KHNYN-2 for wild-type HIV-1 are $2.76 \times 10^{-9}$ and $2.20 \times 10^{-6}$, respectively. p-Values for GFP verses KHNYN-1 and KHNYN-2 for HIV-1$_{EnvCpG86-561}$ are $1.50 \times 10^{-3}$ and $1.51 \times 10^{-3}$, respectively. Gag expression in the media as well as Gag, Hsp90, Env, Actin, KHNYN-FLAG and GFP-FLAG expression in the cell lysates was detected using quantitative immunoblotting (F).

DOI: https://doi.org/10.7554/eLife.46767.004

The following figure supplement is available for figure 2:

**Figure supplement 1.** HIV-1$_{EnvCpG86-561}$ contains 36 introduced CpG dinucleotides.

DOI: https://doi.org/10.7554/eLife.46767.005

infectivity by KHNYN-1 or KHNYN-2 correlated with decreases in Gag expression, Env expression, and virion production (*Figure 2F*). Overall, KHNYN appeared to selectively inhibit HIV-1$_{EnvCpG86-561}$ infectious virus production.

We then determined whether ZAP is necessary for KHNYN to inhibit HIV-1 with clustered CpG dinucleotides. Control or ZAP knockout cells (*Figure 3A*) were co-transfected with pHIV-1 or pHIV-1$_{EnvCpG86-561}$ and increasing amounts of pKHNYN-1. Wild-type HIV-1 infectious virus production was not affected by ZAP depletion and HIV-1$_{EnvCpG86-561}$ infectivity was restored in ZAP knockout cells (*Figures 3B*, 0 ng of KHNYN-1), confirming that ZAP is necessary to inhibit HIV-1 with CpGs introduced in *env* (*Takata et al., 2017*). At low levels of KHNYN-1 overexpression (such as 62.5 ng), there was no substantial decrease in infectivity for wild-type HIV-1 while HIV-1$_{EnvCpG86-561}$ infectivity was inhibited in a ZAP-dependent manner (*Figures 3B* and *4A*). The decrease in infectivity for HIV-1$_{EnvCpG86-561}$ in control cells transfected with pKHNYN-1 correlated with decreases in Gag expression, Env expression and virion production (*Figure 3C*).

Next, we analyzed how ZAP and KHNYN regulate HIV-1 genomic RNA abundance in cell lysates and media. As expected (*Takata et al., 2017*), HIV-1$_{EnvCpG86-561}$ genomic RNA abundance was decreased in control cells but was similar to wild-type HIV-1 in ZAP knockout cells (*Figure 4B–C*, compare GFP samples). In control cells, 62.5 ng of KHNYN-1 or KHNYN-2 inhibited HIV-1$_{EnvCpG86-561}$ genomic RNA abundance compared to the GFP control (*Figure 4B–C*). Importantly, KHNYN-1 and KHNYN-2 did not affect wild-type HIV-1 genomic RNA levels and did not substantially inhibit HIV-1$_{EnvCpG86-561}$ genomic RNA abundance in ZAP knockout cells. This demonstrates that KHNYN targets HIV-1 RNA containing clustered CpG dinucleotides in a ZAP-dependent manner.

TRIM25 is required for ZAP's antiviral activity, although the mechanism by which it regulates ZAP is unclear (*Li et al., 2017*; *Zheng et al., 2017*). To determine if TRIM25 is necessary for the antiviral activity of KHNYN, 62.5 ng of pKHNYN-1 or pKHNYN-2 was co-transfected with pHIV-1 or pHIV-1$_{EnvCpG86-561}$ in control and TRIM25 knockout cells. Both isoforms of KHNYN inhibited HIV-1$_{EnvCpG86-561}$ much less potently in TRIM25 knockout cells than control cells and had no effect on wild-type HIV-1 in either cell line (*Figure 5A–B*). One possible reason that TRIM25 is necessary for KHNYN antiviral activity could be that it regulates the interaction between ZAP and KHNYN. We pulled down KHNYN-FLAG and western blotted for ZAP in control and TRIM25 knockdown cells (*Figure 5C*). Both isoforms of KHNYN pulled down ZAP in both cell lines, indicating that TRIM25 is not required for the interaction between these proteins. Interestingly, KHNYN also pulled down TRIM25 in control cells (*Figure 5C*). Therefore, we analyzed whether KHNYN interacted with TRIM25 in control and ZAP knockout cells and observed that both isoforms of KHNYN-FLAG immunoprecipitated TRIM25 in the presence and absence of ZAP (*Figure 5D*). In sum, KHNYN requires TRIM25 to inhibit HIV-1 containing clustered CpG dinucleotides, but TRIM25 is not necessary for the interaction

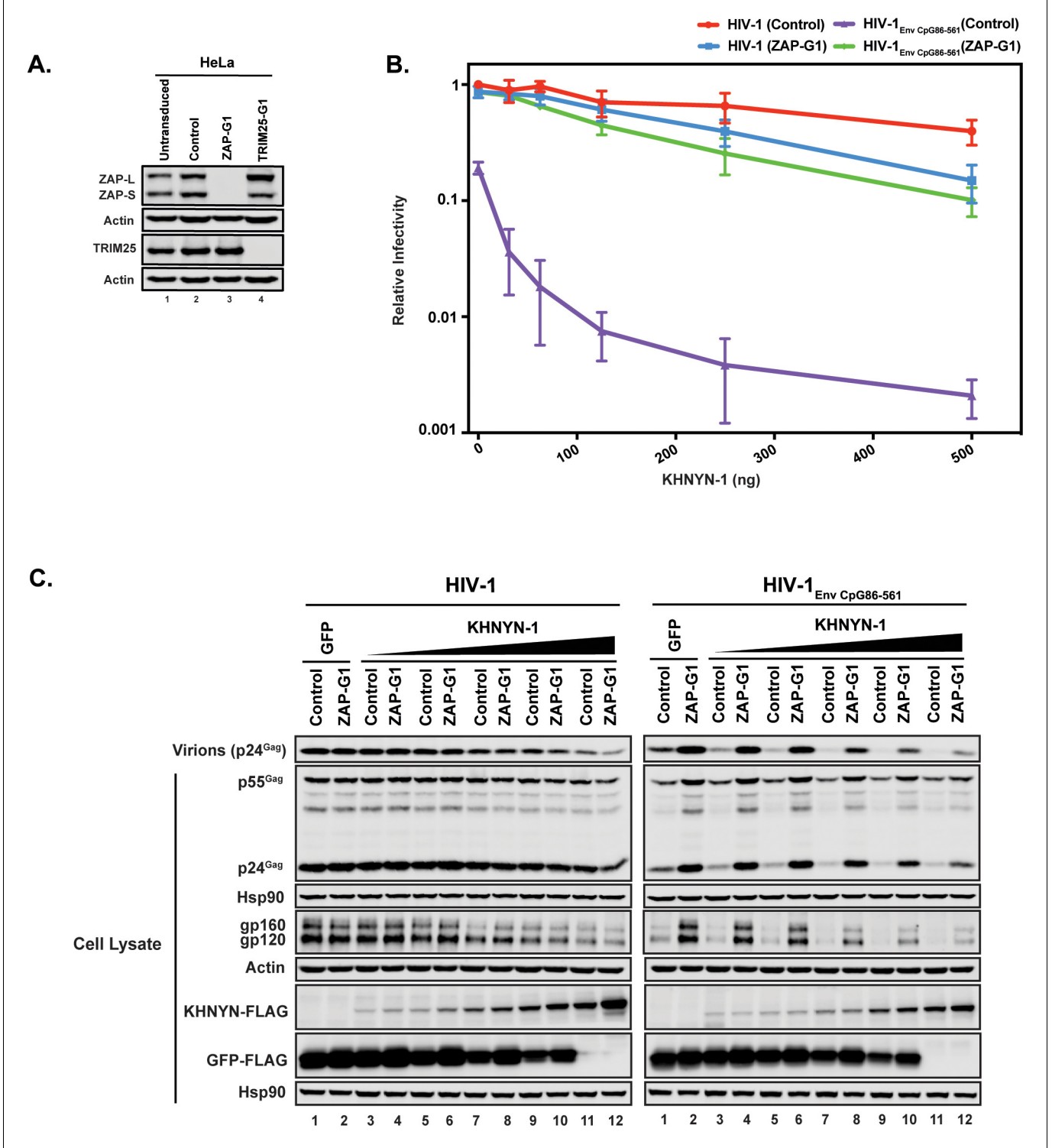

**Figure 3.** ZAP is required for KHNYN to inhibit infectious virion production for HIV-1 with clustered CpG dinucleotides. (A) ZAP, TRIM25 and Actin expression in HeLa cells, HeLa Control CRISPR cells (expressing a guide RNA targeting the *firefly luciferase* gene), HeLa *ZAP* CRISPR guide 1 (ZAP-G1) cells and HeLa *TRIM25* CRISPR guide 1 (TRIM25-G1) cells were detected using quantitative immunoblotting. (B–C) HeLa Control CRISPR cells or ZAP-G1 CRISPR cells were transfected with 500 ng pHIV-1 or pHIV-1$_{EnvCpG86-561}$ and 500 ng of pGFP-FLAG or 31.25 ng, 62.5 ng, 125 ng, 250 ng or 500 ng pKHNYN-1-FLAG plus the amount of pGFP-FLAG required to make 500 ng total. Culture supernatants from the cells were used to infect TZM-bl reporter cells (B). Each point shows the average value of three independent experiments normalized to the value obtained for HeLa Control CRISPR

*Figure 3 continued on next page*

*Figure 3 continued*

cells co-transfected with pHIV-1 and pGFP-FLAG. Data are represented as mean ± SD. Gag expression in the media as well as Gag, Hsp90, Env, Actin, KHNYN-FLAG and GFP-FLAG expression in the cell lysates was detected using quantitative immunoblotting (C).

DOI: https://doi.org/10.7554/eLife.46767.006

between ZAP and KHNYN. Furthermore, ZAP, KHNYN and TRIM25 appear to be in a complex together.

## The KH-like and NYN domains are necessary for KHNYN antiviral activity

As its name implies, KHNYN contains a KH-like domain and a NYN domain (*Figure 6A*). The KH-like domain differs from canonical KH domains due to a potential small metal chelating module containing two cysteines and a histidine inserted into the central region of the domain (*Anantharaman and Aravind, 2006*). Since this has diverged substantially from a standard KH domain, it has also been called a CGIN1 domain and is only known to be present in two other proteins (*Marco and Marín, 2009*). While most KH domains bind nucleic acids (*Nicastro et al., 2015*), the insertion in the KH-like domain in KHNYN may disrupt RNA binding and indicate that it has a different function. To analyze the functional importance of the KH-like domain, we deleted it and found that KHNYN-1ΔKH and KHNYN-2ΔKH had reduced antiviral activity compared to the wild-type protein (*Figure 6B–C*). These mutant proteins localized to the cytoplasm and formed foci that were not present for wild-type KHNYN-1 or KHNYN-2 (*Figure 6—figure supplement 1*).

NYN domains have endonuclease activity and belong to the PIN nuclease domain superfamily. There are at least eight human proteins with a potentially active NYN domain and they have been structurally characterized in several proteins including ZC3H12A and MARF1 (*Matelska et al., 2017*; *Matsushita et al., 2009*; *Nishimura et al., 2018*; *Xu et al., 2012*; *Yao et al., 2018*). These domains contain a negatively charged active site with four aspartic acid residues coordinating a magnesium ion, which activates a water molecule for nucleophilic attack of the phosphodiester group on the target RNA. Mutation of these acidic residues inhibits nuclease activity by disrupting the bonds that directly or indirectly interact with the magnesium ion. ZC3H12A (also known as MCPIP1 and Regnase) is a RNA binding protein that, similar to ZAP, contains a CCCH zinc finger domain and degrades cellular and viral RNAs (*Takeuchi, 2018*). The NYN domain in ZC3H12A has 56% identity to the NYN domain in KHNYN (*Figure 6—figure supplement 2A*). ZC3H12A containing a D141N mutation in the NYN domain had decreased endonuclease activity and did not degrade RNA containing the IL-6 3' UTR (*Matsushita et al., 2009*). MARF1 is required for meiosis and retrotransposon silencing in oocytes and a D426A/D427A mutation inhibited its endoribonuclease activity (*Nishimura et al., 2018*; *Su et al., 2012*). We made these equivalent mutations in KHNYN (*Figure 6A* and *Figure 6—figure supplement 2B*) and tested their ability to inhibit HIV-1$_{EnvCpG86-561}$ gene expression and infectious virus production (*Figure 6B–C*). KHNYN-1 D443N and KHNYN-2 D484N had substantially decreased activity against HIV-1$_{EnvCpG86-561}$. Strikingly, KHNYN-1 D524A/D525A and KHNYN-2 D565A/D566A had no antiviral activity even though they were expressed at similar levels to the wild-type KHNYN. KHNYN proteins with mutations in the NYN domain localized to the cytoplasm, indicating that disruption of this domain's activity did not substantially affect their subcellular localization (*Figure 6—figure supplement 1*).

## KHNYN is necessary for CpG dinucleotides to inhibit HIV-1 RNA and protein expression

To determine if KHNYN is required for CpG dinucleotides to inhibit infectious HIV-1 production, we depleted it using CRISPR-Cas9-mediated genome editing using single-guide RNAs (sgRNAs) targeting two independent sequences in *KHNYN* (*Figure 7A-C* = sgRNA 1, *Figure 7—figure supplement 1* = sgRNA 2). We were unable to identify an antibody that detected endogenous KHNYN. However, FLAG-tagged KHNYN-1 and KHNYN-2 were depleted in the bulk population of cells expressing each sgRNA as well as two clonal cell lines from each bulk population (*Figure 7A* and *Figure 7—figure supplement 1A*). Importantly, when the CRISPR PAM sequence was mutated in the KHNYN plasmids, KHNYN-1 and KHNYN-2 were no longer depleted. The relevant region of *KHNYN* was

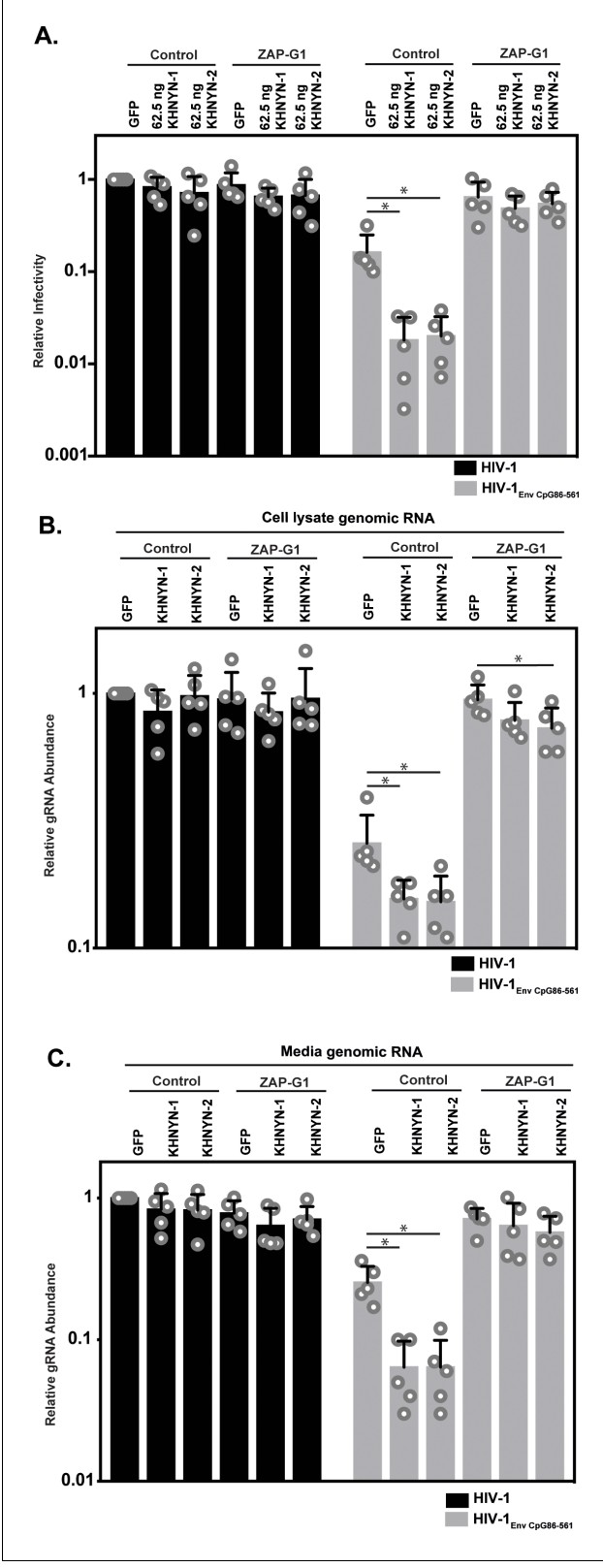

**Figure 4.** ZAP is required for KHNYN to inhibit genomic RNA abundance for HIV-1 with clustered CpG dinucleotides. (**A–C**) HeLa Control CRISPR cells or ZAP-G1 CRISPR cells were transfected with 500 ng pHIV-1 or pHIV-1$_{EnvCpG86-561}$ and 500 ng of pGFP-FLAG or 62.5 ng pKHNYN-1/2-FLAG plus 437.5 ng of pGFP-FLAG. Culture supernatants from the cells were used to infect TZM-bl reporter cells (**A**). The bar chart shows the average value of

*Figure 4 continued on next page*

*Figure 4 continued*

five independent experiments normalized to the value obtained for HeLa Control CRISPR cells co-transfected with pHIV-1 and pGFP-GFP. Data are represented as mean ± SD. *p<0.05 as determined by a two-tailed unpaired t-test. p-values for GFP verses KHNYN-1 and KHNYN-2 for HIV-1$_{EnvCpG86-561}$ in Control cells are $6.78 \times 10^{-3}$ and $7.20 \times 10^{-3}$, respectively. p-Values for GFP verses KHNYN-1 and KHNYN-2 for HIV-1$_{EnvCpG86-561}$ in ZAP-G1 cells are $3.22 \times 10^{-1}$ and $5.33 \times 10^{-1}$, respectively. RNA was extracted from cell lysates (B) and media (C) and genomic RNA (gRNA) abundance was quantified by qRT-PCR. The bar charts show the average value of five independent experiments normalized to the value obtained for HeLa Control CRISPR cells co-transfected with pHIV-1 and pGFP-GFP. Data are represented as mean ± SD. *p<0.05 as determined by a two-tailed unpaired t-test. For HIV-1$_{EnvCpG86-561}$ genomic RNA in Control cell lysates, the GFP verses KHNYN-1 and KHNYN-2 p-values are $2.14 \times 10^{-2}$ and $2.30 \times 10^{-2}$, respectively. For HIV-1$_{EnvCpG86-561}$ genomic RNA in ZAP-G1 cell lysates, the GFP verses KHNYN-1 and KHNYN-2 p-values are $1.01 \times 10^{-1}$ and $4.33 \times 10^{-2}$, respectively. For HIV-1$_{EnvCpG86-561}$ genomic RNA in Control cell media, p-values for GFP verses KHNYN-1 and KHNYN-2 are $8.97 \times 10^{-4}$ and $9.38 \times 10^{-4}$, respectively. For HIV-1$_{EnvCpG86-561}$ genomic RNA in ZAP-G1 cell media, p-values for GFP verses KHNYN-1 and KHNYN-2 are $6.09 \times 10^{-1}$ and $1.87 \times 10^{-1}$, respectively.

DOI: https://doi.org/10.7554/eLife.46767.007

also sequenced to identify the insertion or deletion in the bulk populations as well as the clones and genetic alterations that inactive the protein were identified in each (*Figure 7—figure supplement 2*). These KHNYN CRISPR cells were then transfected with pHIV-1 or pHIV-1$_{EnvCpG86-561}$. In the KHNYN CRISPR cells, the CpG dinucleotides in HIV-1$_{EnvCpG86-561}$ no longer inhibited infectious virus production, Gag expression or Env expression (*Figure 7B–C*, *Figure 7—figure supplement 1B–C*). Wild-type HIV-1 Gag expression, Env expression and infectious virus production was not altered in the KHNYN CRISPR cells compared to the control cells. Overall, the KHNYN CRISPR cells phenocopied the ZAP CRISPR cells (*Figure 7B–C*).

We also analyzed the effect of KHNYN depletion on murine leukemia virus (MLV). While most retroviruses are suppressed in CpG abundance, the degree of this suppression varies between the different genera (*Berkhout et al., 2002*). HIV-1$_{NL4-3}$ is highly suppressed (9 CpGs/kb; 0.2 observed/expected), which is conserved in HIV-1 (*Berkhout et al., 2002*; *Kypr et al., 1989*; *Shpaer and Mullins, 1990*). However, the CpG abundance in MLV is much less suppressed (35 CpGs/kb; 0.5 observed/expected) and ZAP was initially identified as an antiviral protein based on its ability to bind MLV RNA and target it for degradation (*Gao et al., 2002*; *Guo et al., 2004*; *Guo et al., 2007*). To determine if KHNYN inhibits MLV, control, ZAP and KHNYN CRISPR cells were co-transfected with pMLV, p2.87 Vpu (which encodes a highly active HIV-1 Vpu to counteract endogenous Tetherin expression in these cells [*Neil et al., 2008*; *Pickering et al., 2014*]) and pGFP. MLV Gag expression and virion production were measured by immunoblotting. Since ZAP is a type I interferon-stimulated gene (*Shaw et al., 2017*), MLV Gag expression and virion production were also analyzed after type I interferon treatment. There was a small but reproducible increase in MLV Gag expression and virion production in the ZAP and KHNYN CRISPR cells in the absence of type I interferon (*Figure 7D*). However, after type I interferon treatment, MLV virion production was decreased to almost undetectable levels in the control CRISPR cells but was substantially higher in the ZAP and KHNYN CRISPR cells (*Figure 7E*).

The Sindbis virus genome is not substantially depleted in CpG dinucleotides (58 CpGs/kb; 0.9 observed/expected) and is restricted by ZAP (*Bick et al., 2003*). However, unlike retroviruses, the predominant ZAP antiviral activity for alphaviruses is to inhibit viral RNA translation, although there may be an additional effect on RNA stability (*Bick et al., 2003*; *Kozaki et al., 2015*). As expected (*Bick et al., 2003*; *Kozaki et al., 2015*; *Li et al., 2017*; *Zheng et al., 2017*), depletion of TRIM25 and ZAP substantially increased Sindbis virus replication (*Figure 7—figure supplement 3*). In contrast, there was no substantial increase in Sindbis virus replication in the KHNYN CRISPR cells. Thus, KHNYN appears to be required for the restriction of retroviral genomes, but not all ZAP-sensitive RNA viruses.

We then analyzed HIV-1 genomic RNA abundance in the KHNYN CRISPR cells. Similar to HIV-1 protein expression and infectivity, HIV-1$_{EnvCpG86-561}$ genomic RNA abundance was similar in the cell lysate and media to wild-type HIV-1 (*Figure 8A and C*), indicating that the CpG dinucleotides no longer inhibited RNA abundance. As expected, *nef* mRNA abundance was not affected by the

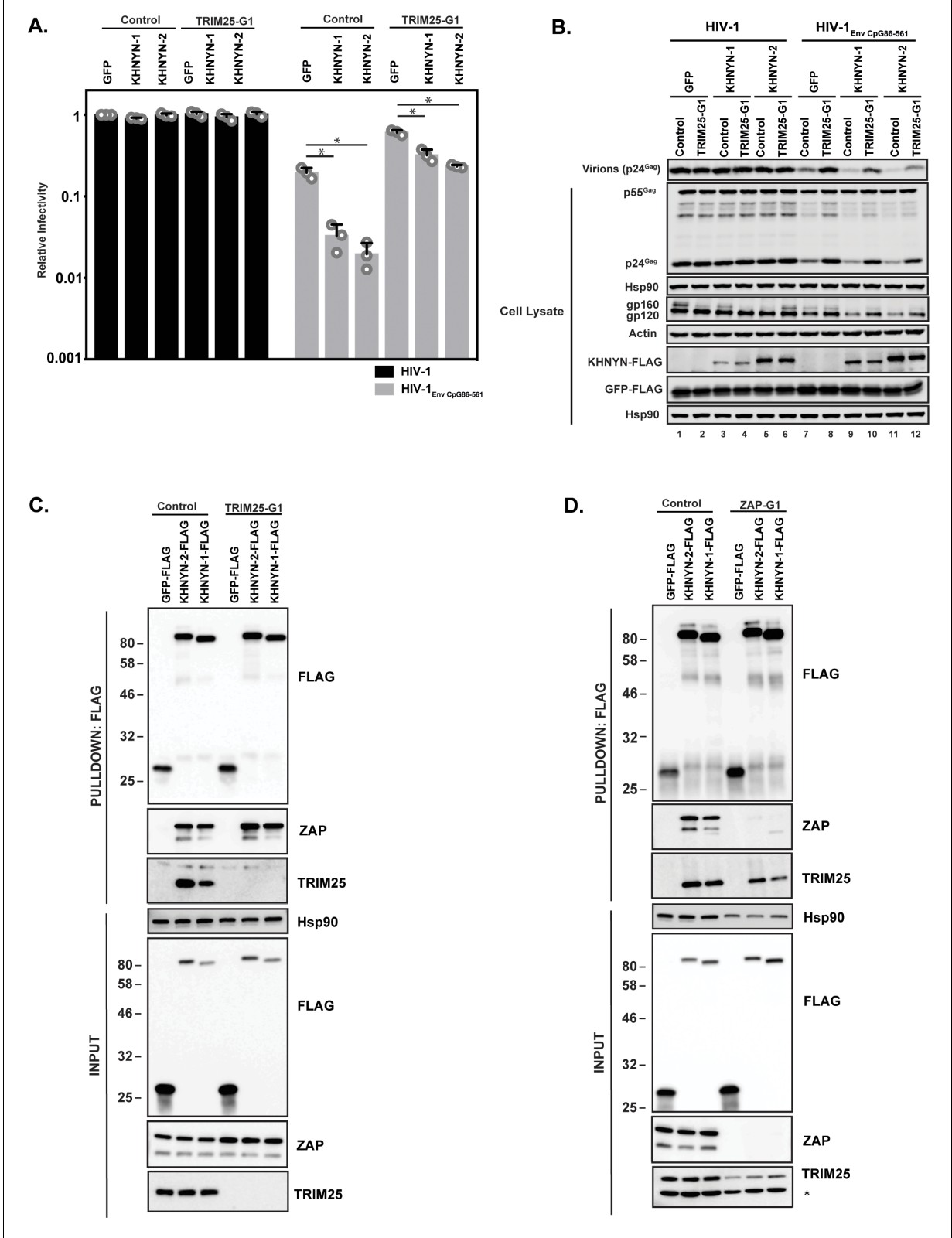

**Figure 5.** TRIM25 is required for KHNYN to inhibit HIV-1 with clustered CpG dinucleotides. (A–B) HeLa Control CRISPR cells or TRIM25-G1 CRISPR cells were transfected with 500 ng pHIV-1 or pHIV-1$_{EnvCpG86-561}$ and 500 ng of pGFP-FLAG or 62.5 ng pKHNYN-1-FLAG or pKHNYN-2-FLAG plus 437.5 ng of pGFP-FLAG. Culture supernatants from the cells were used to infect TZM-bl reporter cells (A). The bar charts show the average value of three independent experiments normalized to the value obtained for HeLa Control CRISPR cells co-transfected with pHIV-1 and pGFP-FLAG. Data are

*Figure 5 continued on next page*

*Figure 5 continued*

represented as mean ± SD. *p<0.05 as determined by a two-tailed unpaired t-test. p-values for GFP verses KHNYN-1 and KHNYN-2 for HIV-1$_{EnvCpG86-561}$ in Control cells are $8.95 \times 10^{-4}$ and $5.42 \times 10^{-4}$, respectively. p-values for GFP verses KHNYN-1 and KHNYN-2 for HIV-1$_{EnvCpG86-561}$ in TRIM25-G1 CRISPR cells are $1.78 \times 10^{-3}$ and $1.01 \times 10^{-4}$, respectively. Gag expression in the media as well as Gag, Hsp90, Env, Actin, KHNYN-FLAG and GFP-FLAG expression in the cell lysates was detected using quantitative immunoblotting (B). (C) Lysates of Control and TRIM25 CRISPR HEK293T cells transfected with pGFP-FLAG, pKHNYN-1-FLAG or pKHNYN-2-FLAG were immunoprecipitated with an anti-FLAG antibody. Post-nuclear supernatants and immunoprecipitation samples were analyzed by immunoblotting for HSP90, KHNYN-FLAG, TRIM25 and ZAP. The blots are representative of two independent experiments. (D) Lysates of Control and ZAP CRISPR HEK293T cells transfected with pGFP-FLAG, pKHNYN-1-FLAG or pKHNYN-2-FLAG were immunoprecipitated with an anti-FLAG antibody. Post-nuclear supernatants and immunoprecipitation samples were analyzed by immunoblotting for HSP90, KHNYN-FLAG, TRIM25 and ZAP. * indicates a non-specific band. The blots are representative of two independent experiments.

DOI: https://doi.org/10.7554/eLife.46767.008

introduction of CpG dinucleotides in *env* or by KHNYN depletion since it is a fully spliced mRNA that does not contain the introduced CpGs (*Figure 8B*). The wild-type HIV-1 genomic RNA abundance was not altered in the KHNYN CRISPR cells compared to the control cells, further showing the specific effect of KHNYN for viral RNA containing CpG dinucleotides.

To determine the specificity of the KHNYN knockdown, we titrated CRISPR-resistant pKHNYN-1 or pKHNYN-2 into the KHNYN CRISPR cells. Even very low levels of KHNYN-1 or KHNYN-2 restored selective inhibition of HIV-1$_{EnvCpG86-561}$ in these cells (*Figure 9A–B*) and KHNYN-1 was consistently slightly more active than KHNYN-2. This shows that both isoforms are capable of inhibiting infectious virus production of HIV-1 containing clustered CpG dinucleotides. We also analyzed whether KHNYN with the KH-like domain deleted or the putative catalytic mutations in the NYN domain could inhibit HIV-1$_{EnvCpG86-561}$ infectious virus production in the CRISPR cells. Expression of 31.25 ng of KHNYN-1 in the KHNYN CRISPR cells inhibited HIV-1$_{EnvCpG86-561}$ (*Figure 9C–D*) and all of the mutations substantially reduced KHNYN antiviral activity. In sum, endogenous KHNYN is required for CpG dinucleotides to inhibit HIV-1 infectious virus production.

## Discussion

Several members of the CCCH zinc finger domain protein family target viral and/or cellular mRNAs for degradation (*Fu and Blackshear, 2017*). For example, ZC3H12A degrades pro-inflammatory cytokine mRNAs and also inhibits the replication of several viruses, including HIV-1 and hepatitis C virus (*Lin et al., 2013*; *Lin et al., 2014*; *Liu et al., 2013*; *Matsushita et al., 2009*). It contains a CCCH zinc finger domain as well as a NYN endonuclease domain, which allows it to degrade specific RNAs (*Matsushita et al., 2009*; *Xu et al., 2012*). ZAP has four CCCH zinc finger domains and specifically interacts with CpG dinucleotides in RNA (*Gao et al., 2002*; *Guo et al., 2004*; *Takata et al., 2017*). However, it does not contain nuclease activity. While ZAP has been reported to directly or indirectly interact with components of the 5′−3′ and 3′−5′ degradation pathways including DCP1-DCP2, XRN1, PARN and the exosome, knockdown of several proteins in these pathways did not substantially rescue infectious virus production of HIV-1 containing clustered CpG dinucleotides (*Goodier et al., 2015*; *Guo et al., 2007*; *Takata et al., 2017*; *Zhu et al., 2011*). Therefore, we hypothesized that ZAP may interact with additional unidentified proteins that regulate viral RNA degradation.

Herein, we have identified that KHNYN is an essential ZAP cofactor that inhibits HIV-1 gene expression and infectious virus production when the viral RNA contains clustered CpG dinucleotides. KHNYN overexpression inhibits genomic RNA abundance, Gag expression, Env expression and infectious virus production for HIV-1 containing clustered CpG dinucleotides. This activity requires ZAP and TRIM25. Furthermore, depletion of KHNYN using CRISPR-Cas9 specifically increased genomic RNA abundance, Gag expression, Env expression and infectious virus production for HIV-1 containing clustered CpG dinucleotides. This indicates that KHNYN is essential for CpG dinucleotides to inhibit infectious virus production. Similarly, KHNYN depletion increased MLV Gag expression and virion production. However, Sindbis virus replication was not substantially increased in the KHNYN knockout cells. The difference between the requirement for KHNYN to inhibit retroviruses versus Sindbis virus may be because the antiviral effect for retroviruses is predominantly at the level of RNA stability and for alphaviruses it is predominately at the level of translation (*Bick et al., 2003*;

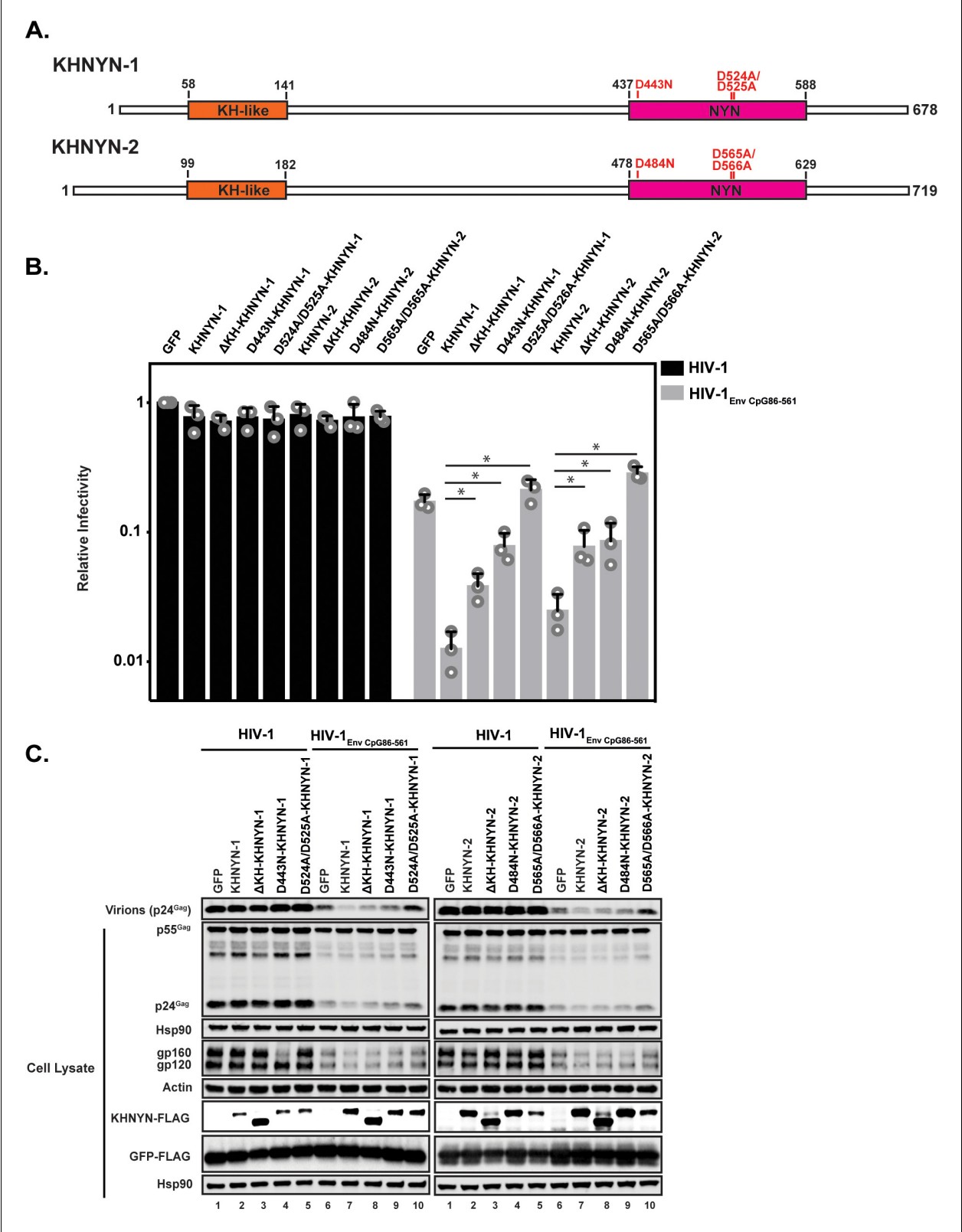

**Figure 6.** The KH and NYN domains are necessary for KHNYN antiviral activity. (**A**) Schematic of KHNYN-1 and KHNYN-2 domains and mutations in the NYN domain. (**B–C**) HeLa cells were transfected with 500 ng pHIV-1 or pHIV-1$_{EnvCpG86-561}$ and either 500 ng of pGFP-FLAG or 62.5 ng pKHNYN-1/2-FLAG and 437.5 ng of pGFP-FLAG. Culture supernatants were used to infect TZM-bl reporter cells to measure infectivity (**B**). The bar charts show the average values of three independent experiments normalized to the value obtained for HeLa cells co-transfected with pHIV-1 and pGFP-FLAG. Data

*Figure 6 continued on next page*

Figure 6 continued

are represented as mean ± SD. *p<0.05 as determined by a two-tailed unpaired t-test. p-values for KHNYN-1 versus ΔKH-KHNYN-1, D443N-KHNYN-1, and D524A/D525A-KHNYN-1 for HIV-1$_{EnvCpG86-561}$ are $1.31 \times 10^{-2}$, $5.22 \times 10^{-3}$, and $1.26 \times 10^{-3}$, respectively. p-values for KHNYN-2 versus ΔKH-KHNYN-2, D443N-KHNYN-2, and D524A/D525A-KHNYN-2 for HIV-1$_{EnvCpG86-561}$ are $2.88 \times 10^{-2}$, $3.26 \times 10^{-2}$, and $2.56 \times 10^{-4}$, respectively. Gag expression in the media as well as Gag, Hsp90, Env, Actin, KHNYN-1/2-FLAG and GFP-FLAG expression in the cell lysates was detected using quantitative immunoblotting (C). See also *Figure 6—figure supplements 1* and *2*.
DOI: https://doi.org/10.7554/eLife.46767.009
The following figure supplements are available for figure 6:

**Figure supplement 1.** KHNYN mutant proteins localize to the cytoplasm.
DOI: https://doi.org/10.7554/eLife.46767.010
**Figure supplement 2.** NYN domain alignment.
DOI: https://doi.org/10.7554/eLife.46767.011

*Gao et al., 2002*; *Guo et al., 2004*; *Guo et al., 2007*; *MacDonald et al., 2007*). A mechanistic explanation for why the major antiviral effect of ZAP appears to be promoting RNA degradation for some viruses and inhibiting translation for other viruses remains unclear, although ZAP has been reported to inhibit translation initiation by interfering with the interaction between eIF4A and eIF4G (*Zhu et al., 2012*). Therefore, KHNYN may not be required for ZAP to inhibit translation.

We hypothesize that a complex containing ZAP and KHNYN binds HIV-1 CpG-containing RNA. ZAP and KHNYN could directly interact to form a heterodimer or there could be other factors mediating this interaction. The interaction between ZAP and KHNYN has been detected using several different assays including yeast-two-hybrid, co-immunoprecipitation, affinity purification–mass spectrometry (*Huttlin et al., 2017*) and BioID (*Youn et al., 2018*). If there is an unknown factor mediating this interaction, it would have to present in the yeast-two-hybrid assay. It remains unclear how TRIM25 regulates ZAP, but it is not required for ZAP and KHNYN to interact. Interestingly, TRIM25 co-immunoprecipitates with KHNYN and the ZAP antiviral complex may simultaneously consist of all three proteins. ZAP and TRIM25 are interferon-stimulated genes while KHNYN is not induced by interferon in human cells (*Shaw et al., 2017*). Whether KHNYN is regulated by type I interferons or viral infection in a different way, such as post-translational modification, is not known.

The zinc finger RNA binding domains in ZAP could target KHNYN to CpG regions in viral RNA. This would allow the endonuclease domain in KHNYN to cleave this RNA, thereby inhibiting viral RNA abundance. Conceptually, the ZAP-KHNYN complex could function similarly to ZC3H12A, but with the RNA binding and endonuclease domains divided between the two proteins. The NYN domain in KHNYN could cleave HIV-1 RNA containing CpG dinucleotides similar to how ZC3H12A cleaves a specific site in the 3' UTR of the IL-6 mRNA (*Matsushita et al., 2009*). While we do not yet have evidence that the NYN domain in KHNYN is an active endonuclease domain, it is highly conserved with the active NYN domain in ZC3H12A and is required for KHNYN antiviral activity. Strikingly, mutation of two conserved aspartic acid residues in the NYN domain predicted to coordinate a magnesium ion necessary for nucleophilic attack of the target RNA eliminated KHNYN antiviral activity. However, biochemical and structural studies will be necessary to determine the specific nature of the interaction between ZAP, KHNYN, TRIM25 and RNA and how these interactions promote viral RNA degradation.

An increasingly common theme for RNA decay is that endonucleic and exonucleic degradation pathways work together to fully degrade RNAs. For example, nonsense-mediated decay (NMD) targets mRNAs that do not efficiently terminate translation at the stop codon and uses up to four mechanisms to degrade these mRNAs: endonucleic cleavage, deadenylation, decapping and exonucleic degradation (*Lykke-Andersen and Jensen, 2015*). In this pathway, the endonuclease SMG6 interacts with the core regulatory protein UPF1 and cleaves mRNA near a premature termination codon. The 5' and 3' cleavage fragments are then degraded by the 5'−3' exonuclease XRN1 and the 3'−5' exonuclease exosome complex. The CCR4–NOT deadenylase complex and DCP1-DCP2 decapping complex are recruited by proteins in the NMD complex including UPF1, SMG5, SMG7 and PNRC2. Similarly, the 5'−3' and 3'−5' degradation pathway components previously shown to interact with ZAP could work in conjunction with KHNYN-mediated endonucleic decay (*Goodier et al., 2015*; *Guo et al., 2007*; *Zhu et al., 2011*). In this model, KHNYN would initiate cleavage of the viral RNA and the exonucleic pathways would then degrade the resulting RNA

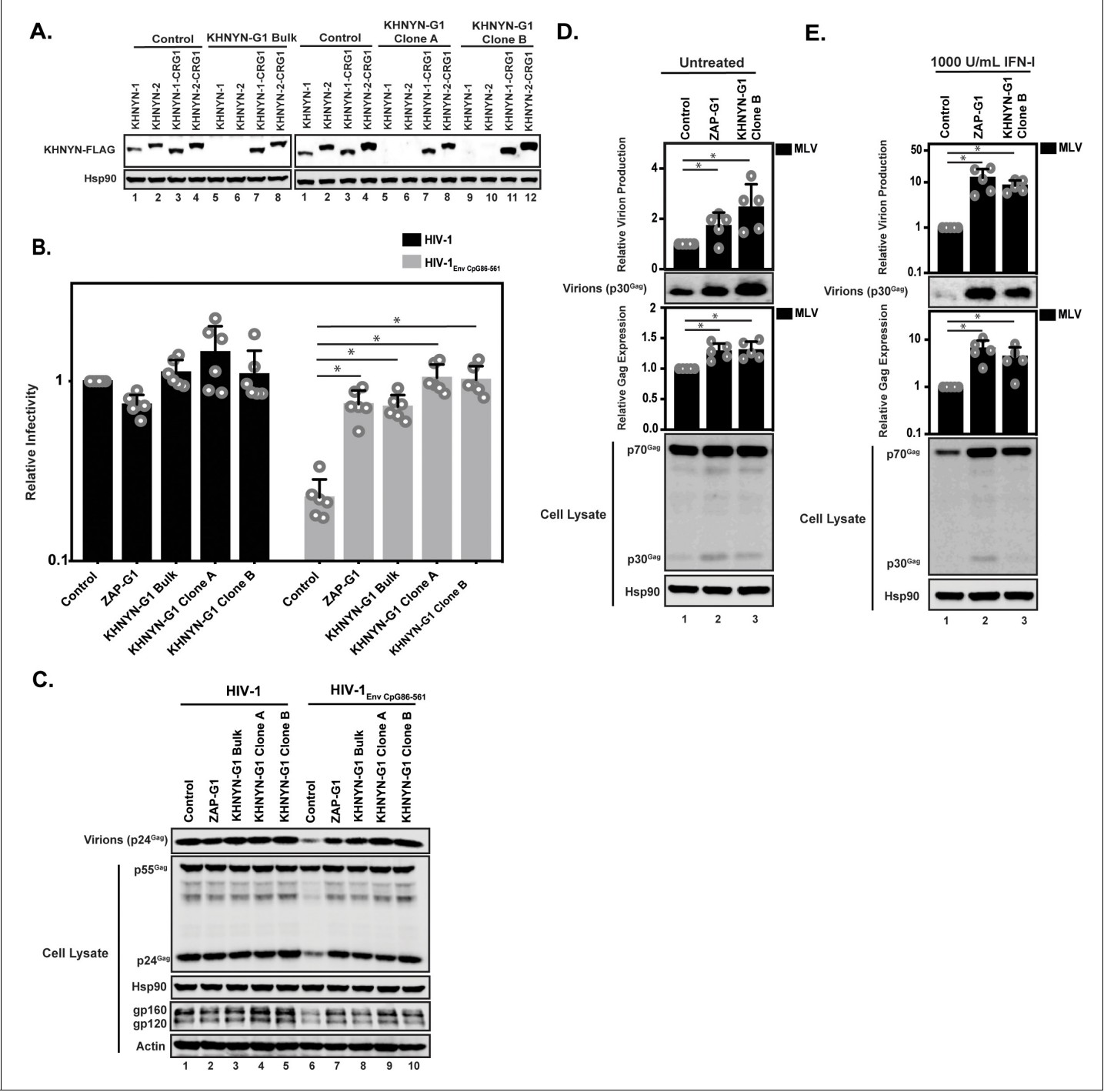

**Figure 7.** KHNYN depletion increases infectious virus production for HIV-1 with clustered CpG dinucleotides and MLV Gag expression. (A) 100 ng pKHNYN-1-FLAG, pKHNYN-2-FLAG, pKHNYN-1-FLAG with a mutation in the PAM that prevents it from being targeted by *KHNYN* guide 1 (pKHNYN-1-CRG1-FLAG) or pKHNYN-2-CRG1-FLAG plus 400 ng of pGFP were transfected into HeLa Control CRISPR cells, *KHNYN* guide 1 (KHNYN-G1) CRISPR bulk cells or KHNYN-G1 CRISPR single cell clones A and B. KHNYN-FLAG abundance was measured by quantitative western blotting. (B–C) HeLa Control CRISPR cells, ZAP-G1 CRISPR cells, KHNYN guide 1 (KHNYN-G1) CRISPR bulk cell population or single cell clones A and B were transfected with 500 ng pHIV-1 or pHIV-1$_{EnvCpG86-561}$ and 500 ng pGFP. Culture supernatants were used to infect TZM-bl reporter cells to measure infectivity (B). The bar charts show the average value of six independent experiments normalized to the value obtained for HeLa Control CRISPR cells co-transfected with pHIV-1 and pGFP. Data are represented as mean ± SD. *p<0.05 as determined by a two-tailed unpaired t-test. p-values for HIV-1$_{EnvCpG86-561}$ in Control cells versus ZAP-G1, KHNYN-G1 Bulk, KHNYN-G1 Clone A, and KHNYN-G1 Clone B CRISPR cells are $8.23 \times 10^{-6}$, $2.81 \times 10^{-6}$, $1.60 \times 10^{-6}$, and $2.24 \times 10^{-6}$, respectively. Gag expression in the media as well as Gag, Hsp90, Env, and Actin expression in the cell lysates was detected using

*Figure 7 continued on next page*

Figure 7 continued

quantitative immunoblotting (C). (D–E) HeLa Control CRISPR cells, ZAP-G1 CRISPR cells and *KHNYN* G1 CRISPR clone B were transfected with 650 ng pMLV, 250ng p2.87 Vpu and 100 ng pGFP. Gag expression in the media (D) as well as Gag and Hsp90 expression in the cell lysates (E) was detected using quantitative immunoblotting. The bar charts show the average value of five independent experiments normalized to the value obtained for HeLa Control CRISPR cells. Data are represented as mean ± SD. *p<0.05 as determined by a two-tailed unpaired t-test. p-values for virions from Control cells versus ZAP-G1 and KHNYN-G1 Clone B CRISPR cells without type I IFN are $1.28 \times 10^{-3}$ and $1.50 \times 10^{-3}$, respectively, and $4.17 \times 10^{-3}$ and $1.92 \times 10^{-2}$ with type I IFN treatment. p-values for the cell lysates from Control cells versus ZAP-G1 and KHNYN-G1 Clone B CRISPR cells are $1.78 \times 10^{-2}$ and $8.01 \times 10^{-3}$ without type I interferon, respectively, and $6.37 \times 10^{-2}$ and $1.73 \times 10^{-4}$ with type I interferon treatment.

DOI: https://doi.org/10.7554/eLife.46767.012

The following figure supplements are available for figure 7:

**Figure supplement 1.** KHNYN is necessary for CpG dinucleotides to inhibit HIV-1 infectious virus production.

DOI: https://doi.org/10.7554/eLife.46767.013

**Figure supplement 2.** CRISPR-Cas9-induced mutations in *KHNYN*.

DOI: https://doi.org/10.7554/eLife.46767.014

**Figure supplement 3.** KHNYN depletion does not substantially increase Sindbis virus replication.

DOI: https://doi.org/10.7554/eLife.46767.015

fragments. Identifying the full complement of ZAP-interacting factors and characterizing how these target viral RNAs for degradation will be an exciting area of future investigation.

Another important area of future research will be to determine how KHNYN and other cellular proteins that contain an NYN endonuclease domain inhibit the replication of different viruses in different cell types with and without interferon treatment. The interferon-stimulated gene N4BP1, which is a KHNYN paralog (*Anantharaman and Aravind, 2006*), was recently identified to genetically interact with ZAP in a CRISPR-based screen to identify interferon-induced antiviral proteins targeting HIV-1 (*OhAinle et al., 2018*). In the monocytic THP-1 cell line, depletion of N4BP1 led to a small increase in wild-type HIV-1 replication. However, N4BP1 depletion did not inhibit replication of the alphavirus Semliki Forest virus, indicating that it may have a virus specific effect. While ZAP inhibits a range of viruses in different cell types, it remains unknown whether its cofactor requirements are cell type dependent. In this study, we have analyzed the antiviral activity of ZAP and KHNYN on HIV-1 and MLV in HeLa cells, but the role of NYN domain-containing proteins in targeting viral RNAs for degradation may be an important component of the antiviral innate immune response in a variety of cell types. It will also be interesting to determine if proteins containing an endonuclease domain other than KHNYN interact with CCCH zinc finger proteins to mediate antiviral activity. There are 57 human CCCH zinc finger proteins (*Fu and Blackshear, 2017*). At least 15 of these proteins are known to promote RNA decay and, including ZAP, six human CCCH zinc finger proteins are antiviral (*Fu and Blackshear, 2017*). Identifying the full complement of CCCH zinc finger proteins that inhibit viral replication and determining whether they require proteins containing endonuclease domains such as KHNYN or N4BP1 for this activity will increase our understanding of antiviral responses targeting viral RNA.

## Materials and methods

### Plasmids

All primers were ordered from Eurofins. Polymerase chain reactions (PCR) for cloning steps were performed with Phusion High Fidelity polymerase (New England Biolabs). KHNYN-2 (NM_001290256) was synthesized by GenScript. KHNYN-1 was cloned by amplifying the nucleotides 123–2157 from KHNYN-2 and sub-cloning the PCR product into the pcDNA3.1 (+) backbone using the HindIII site in the vector and SbfI site in the KHNYN open reading frame. KHNYN-1 ΔKH, D443N, D524A/D525A, -CRG1, -CRG2 and KHNYN-2 ΔKH, D484N, D565A/D566A, -CRG1, -CRG2 were generated via overlap extension PCR and subsequently sub-cloning the PCR product into the pcDNA3.1 (+) backbone as described above. pGFP-FLAG was cloned by amplifying GFP from pcDNA3.1-GFP (*Swanson et al., 2010*) and cloning it into pcDNA3.1. Diagnostic restriction enzyme digestion and DNA sequencing (Eurofins, Genewiz) was used to ensure the correct identity of modified sequences inserted into plasmids. pHIV-1$_{NL4-3}$ contains the HIV-1 provirus in the pGL4 vector

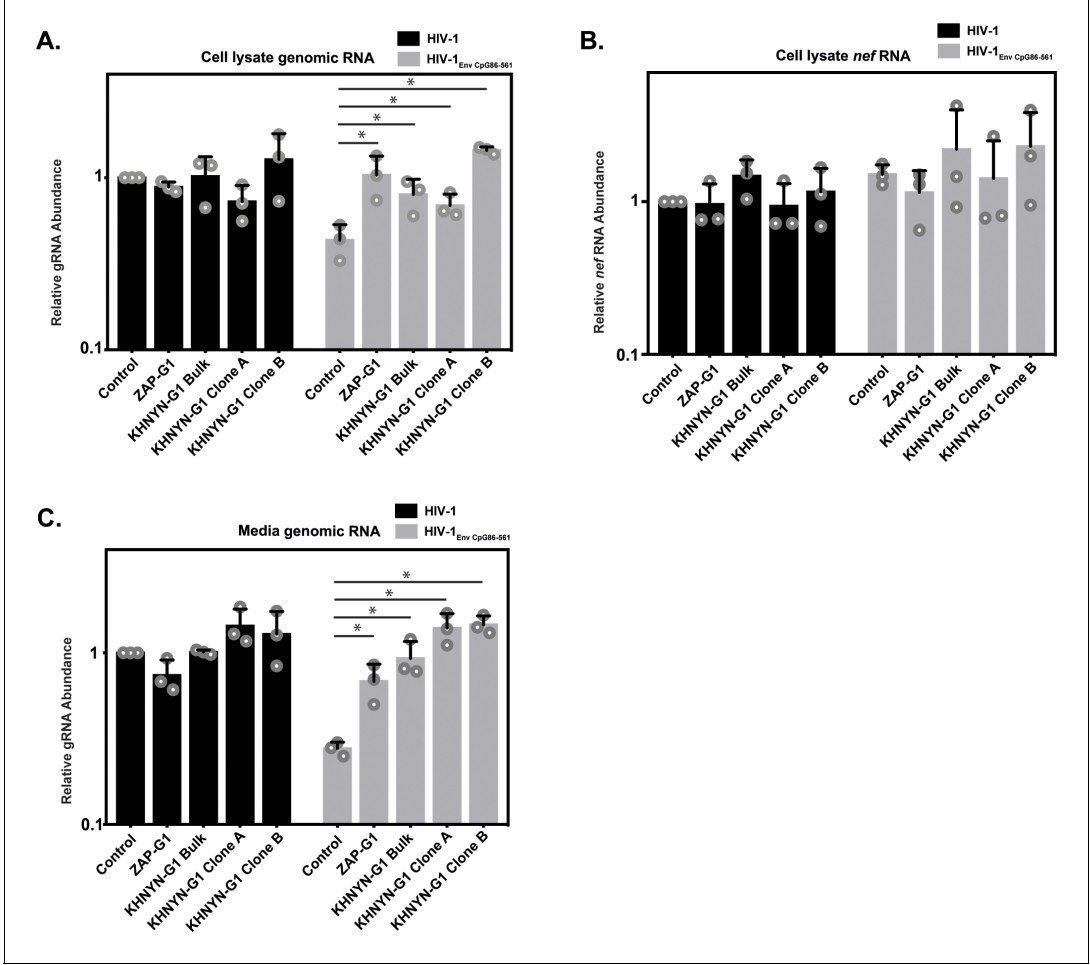

**Figure 8.** KHNYN is necessary for CpG dinucleotides to inhibit HIV-1 genomic RNA expression. (**A–C**) HeLa Control CRISPR cells, ZAP-G1 CRISPR cells, KHNYN-G1 CRISPR bulk cells or two independent clones were transfected with 500 ng pHIV-1 or pHIV-1$_{EnvCpG86-561}$ and 500 ng pGFP-FLAG. RNA was extracted from cell lysates (**A–B**) and media (**C**). Genomic RNA (gRNA) (**A, C**) or *nef* mRNA (**B**) abundance was quantified by qRT-PCR. The bar charts show the average value of three independent experiments normalized to the value obtained for HeLa Control CRISPR cells co-transfected with pHIV-1 and pGFP-FLAG. Data are represented as mean ± SD. *p<0.05 as determined by a two-tailed unpaired t-test. p-values for HIV-1$_{EnvCpG86-561}$ genomic RNA in Control cell versus ZAP-G1, KHNYN-G1 Bulk, KHNYN-G1 Clone A, and KHNYN-G1 Clone B CRISPR cell lysates are $2.98 \times 10^{-2}$, $3.70 \times 10^{-2}$, $4.26 \times 10^{-2}$, and $1.30 \times 10^{-4}$, respectively. p-Values for HIV-1$_{EnvCpG86-561}$ *nef* mRNA in Control cell versus ZAP-G1, KHNYN-G1 Bulk, KHNYN-G1 Clone A, and KHNYN-G1 Clone B CRISPR cell lysates are 0.29, 0.54, 0.90 and 0.42, respectively. p-Values for HIV-1$_{EnvCpG86-561}$ genomic RNA in Control cell versus ZAP-G1, KHNYN-G1 Bulk, KHNYN-G1 Clone A, and KHNYN-G1 Clone B CRISPR cell media are $1.65 \times 10^{-2}$, $8.64 \times 10^{-3}$, $2.82 \times 10^{-3}$, and $3.54 \times 10^{-4}$, respectively.

DOI: https://doi.org/10.7554/eLife.46767.016

(**Antzin-Anduetza et al., 2017**). To generate HIV-1$_{EnvCpG86-561}$, we synthesized a HIV-1$_{NL4-3}$ EcoRI/StuI DNA fragment with synonymous mutations that inserted 36 CpG dinucleotides into *env* (**Figure 2—figure supplement 1**). This DNA fragment was digested with *EcoRI* and *StuI* and inserted into the corresponding sites of pHIV-1$_{NL4-3}$. pMLV is pNCS, which contains the Moloney murine leukemia virus provirus DNA (**Yueh and Goff, 2003**). pCR3.1 2.87 Vpu contains the HA-tagged codon-optimized HIV-1 Vpu 2.87 (**Pickering et al., 2014**).

## Cell lines

HeLa and HEK293T cells were obtained from the ATCC and were maintained in high glucose DMEM supplemented with GlutaMAX, 10% fetal bovine serum, 20 µg/mL gentamicin or 100 U/ml penicillin and 100 µg/ml streptomycin and incubated with 5% $CO_2$ at 37˚C. BHK-21 cells were obtained from ATCC and were maintained in GMEM supplemented with 10% fetal bovine serum, 10% tryptose

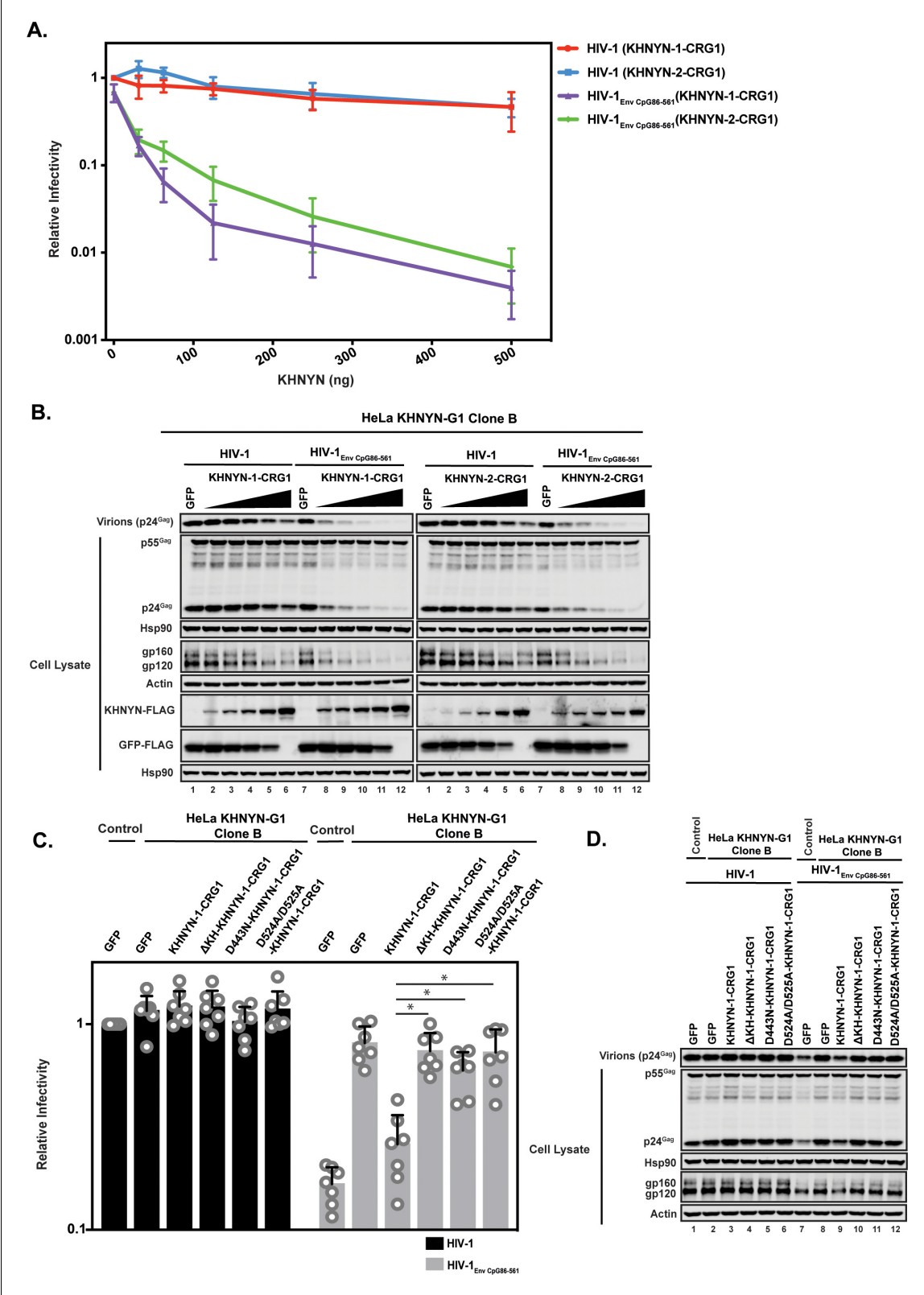

**Figure 9.** Deletion of the KH-like domain or mutations in the NYN domain in KHNYN prevent reconstitution of CpG-dependent inhibition of HIV-1 infectious virus production in KHNYN CRISPR cells. (A–B) HeLa KHNYN-G1 CRISPR clone B cells were transfected with 500 ng pHIV-1 or pHIV-1$_{EnvCpG86-561}$ and 500 ng of pGFP-FLAG or 31.25 ng, 62.5 ng, 125 ng, 250 ng or 500 ng of pKHNYN-1-CRG1-FLAG or pKHNYN-2-CRG1-FLAG plus the amount of pGFP-FLAG required to make 500 ng total. Culture supernatants from the cells were used to infect TZM-bl reporter cells (A). Each point

*Figure 9 continued on next page*

*Figure 9 continued*

shows the average value of three independent experiments normalized to the value obtained for HeLa Control CRISPR cells co-transfected with pHIV-1 and pGFP. Data are represented as mean ± SD. Gag expression in the media as well as Gag, Hsp90, Env, Actin, KHNYN-FLAG and GFP-FLAG expression in the cell lysates was detected using quantitative immunoblotting (**B**). (**C–D**) HeLa KHNYN-G1 CRISPR clone B cells were transfected with 500 ng pHIV-1 or pHIV-1$_{EnvCpG86-561}$, 468.75 ng of pGFP-FLAG and 31.25 ng of pKHNYN-1-CRG1-FLAG expressing either wild-type or mutant proteins. Culture supernatants from the cells were used to infect TZM-bl reporter cells (**C**). Each point shows the average value of seven independent experiments normalized to the value obtained for HeLa Control CRISPR cells co-transfected with pHIV-1 and pGFP. Data are represented as mean ± SD. *$p<0.05$ as determined by a two-tailed unpaired t-test. p-Values for HIV-1$_{EnvCpG86-561}$ with KHNYN-1-CRG1 versus ΔKH-KHNYN-1-CGR1, D443N-KHNYN-1-CGR1 and D524A/D525A-KHNYN-1-CGR1 are $2.56 \times 10^{-5}$, $1.95 \times 10^{-4}$, and $2.12 \times 10^{-4}$, respectively. Gag expression in the media as well as Gag, Hsp90, Env, Actin, KHNYN-FLAG and GFP-FLAG expression in the cell lysates was detected using quantitative immunoblotting (**D**).
DOI: https://doi.org/10.7554/eLife.46767.017

phosphate broth, and penicillin/streptomycin. Their identity has not been authenticated and they are routinely tested for mycoplasma contamination with all tests negative. CRISPR guides targeting the firefly luciferase gene, lacZ gene, human TRIM25, ZAP (also known as ZC3HAV1), and KHNYN genes were cloned into *BsmB*I restriction enzyme sites in the lentiviral vector genome plasmid lentiCRISPRv2 (*Sanjana et al., 2014*). The CRISPR guide sequences are: LacZ-G1: 5'- CGA TTA AGT TGG GTA ACG CC −3', Luciferase-G1: 5'-CTT TAC CGA CGC ACA TAT CG-3', TRIM25-G1: 5'-GAG CCG GTC ACC ACT CCG TG −3', ZAP-G1: 5'- ACT TCC ATC TGC CTT ACC GG −3', KHNYN-G1: 5'-GGG GGT GAG CGT CCT TCC GA-3', KHNYN-G2: 5'-CAG ACA CCG CAA AGC GAT CT-3'. LentiCRISPR vectors encoding guide RNAs targeting KHNYN or LacZ were produced in HEK293T cells seeded in a 10 cm dish and transfected using 100 ul PEI with 8 µg of lentiCRISPRv2-Guide, 8 µg of pCRV1-HIV-Gag Pol (*Neil et al., 2008*) and 4 µg of pCMV-VSV-G (*Neil et al., 2008*). Lentiviral vectors encoding guide RNAs targeting ZAP or TRIM25 were produced by transfecting HEK293T cells seeded in a six-well plate using 10 µl PEI with 0.5 µg pVSV-G (*Fouchier et al., 1997*), 1.0 µg pCMVΔR8.91 (*Zufferey et al., 1997*), and 1.0 µg LentiCRISPRv2-Guide. Virus containing supernatant was harvested 48 hr after transfection, rendered cell-free via filtration through 0.45 µM filters (Millipore) and used to transduce HeLa or HEK293T cells followed by selection in puromycin.

## Yeast two-hybrid screen

The yeast two-hybrid screen was performed by Hybrigenics Services, S.A.S., Paris, France (http://www.hybrigenics-services.com). Full length ZAP-S, ZAP-L and KHNYN-2 were PCR-amplified and cloned into pB27 as a C-terminal fusion to LexA. These constructs were used as a bait to screen a random-primed Induced Macrophages cDNA library constructed into pP6. pB27 and pP6 derive from the original pBTM116 (*Vojtek and Hollenberg, 1995*) and pGADGH (*Bartel and Fields, 1995*) plasmids, respectively. Clones were screened using a mating approach with YHGX13 (Y187 ade2-101::loxP-kanMX-loxP, matα) and L40ΔGal4 (mata) yeast strains as previously described (*Fromont-Racine et al., 1997*). His + colonies were selected on a medium lacking tryptophan, leucine and histidine. Because KHNYN-2 had some autoactivating activity, the selection medium was supplemented with 10 mM 3-Aminotriazol.

The prey fragments of the positive clones were amplified by PCR and sequenced at their 5' and 3' junctions. The resulting sequences were used to identify the corresponding interacting proteins in the GenBank database (NCBI) using a fully automated procedure. A confidence score (PBS, for Predicted Biological Score) was attributed to each interaction as previously described (*Formstecher et al., 2005*). The PBS relies on two different levels of analysis. First, a local score takes into account the redundancy and independency of prey fragments, as well as the distribution of reading frames and stop codons in overlapping fragments. Second, a global score takes into account the interactions found in all the screens performed at Hybrigenics using the same library. This global score represents the probability of an interaction being nonspecific. The scores were divided into six categories: A (highest confidence) to D (lowest confidence) plus category E that demarcates interactions involving highly connected prey domains previously found several times in screens performed on libraries derived from the same organism and category F that indicates highly connected domains that have been confirmed as false-positives. The PBS scores have been shown to positively correlate with the biological significance of interactions (*Rain et al., 2001*; *Wojcik et al., 2002*).

## Transfections

HeLa and HEK293T cells were grown to 70% confluence in six-well plates. HeLa cells were transfected according to the manufacturer's instructions using *Trans*IT-LT1 (Mirus) at the ratio of 3 μL *Trans*IT- LT1 to 1 μg DNA. HEK293T cells were transfected according to the manufacturer's instructions using PEI (1 mg/mL) (Sigma-Aldrich) at the ratio of 4 μL PEI to1 μg DNA. For the HIV-1 experiments, 0.5 μg pHIV-1 and the designated amount of pKHNYN-FLAG, pGFP-FLAG or pGFP (*Swanson et al., 2010*) were transfected for a total of 1 μg DNA. For the MLV experiments, 0.65 μg pMLV, 0.25 μg pCR3.1 2.87 Vpu and 0.10 μg pGFP were transfected. The transfection medium was replaced with fresh medium after a 6 hr incubation (HEK293T) or 24 hr incubation (HeLa).

## Analysis of protein expression by immunoblotting

HeLa cells were seeded on six-well plates and transfected the following day with as described above. Approximately 48 hr post-transfection, HeLa cells were lysed in radioimmunoprecipitation assay (RIPA) buffer (150 mM NaCl, 1 mM EDTA, 1% Triton X-100, 1% sodium deoxycholate, 0.1% SDS, 10 mM Tris–HCl pH 7.5) supplemented with complete protease inhibitor (Roche) and sheared using a 0.25G needle. Media was filtered through a 0.45 μM filter and virions were pelleted for 2 hr at 20,000 x g through a 20% sucrose cushion in phosphate-buffered saline (PBS) solution. The pellet was resuspended in 2X loading buffer (60 mM Tris–HCl pH 6.8, 2% sodium dodecyl sulfate (SDS), 10% glycerol, 10% β-mercaptoethanol, 0.1% bromophenol blue). Cell lysates and virions were resolved by 10% SDS-polyacrylamide gel electrophoresis (PAGE), transferred to a nitrocellulose membrane (GE Healthcare) and blocked in 1% non-fat milk in PBS with 0.1% Tween 20 (Fischer Bioreagents). Primary antibodies were incubated for 2 hr at room temperature. After washing in PBS, blots were incubated for 1 hr with the appropriate secondary antibody. Bound antibodies were visualized on the LI-COR (Odyssey Fc) measuring the immunofluorescence or using Amersham ECL Prime Western Blotting Detection reagent (GE Lifesciences) for HRP-linked antibodies using an ImageQuant (LAS4000 Mini). For the co-immunoprecipitation experiments, lysates were resolved using precast Mini-PROTEAN TGX gels 8–16% gradient gels (Bio-Rad) and transferred to nitrocellulose membranes (Bio-Rad). Antibodies used in study were 1:50 HIV- one anti-p24Gag (*Chesebro et al., 1992*), 1:3000 anti-HIV-1 gp160/120 Rabbit (ADP421; Centralized Facility for AIDS Reagents (CFAR)), 1:10,0000 anti-HSP90 (sc7947, Santa Cruz Biotechnology), 1:5000 anti-HSP90 Rabbit (GeneTex, GTX109753), 1:4000 anti-HSP90 Mouse (SantaCruz, sc-515081), 1:5000 anti-ZAP (Abcam, ab154680), 1:1000 anti-β-Actin Mouse (Sigma, A2228), 1:2500 anti-DYKDDDDK (Rabbit) (Cell Signaling, 14793), 1:2500 anti-FLAG (Mouse) (Sigma, F1804), 1:2500 anti-FLAG (Rabbit) (Sigma, F7425), 1:10000 anti-TRIM25 (Abcam, ab167154), 1:10000 anti-MLV p30 (Rat) (ATCC CRL-1912) (*Chesebro et al., 1983*), 1:10,000 Dylight 800- conjugated secondary antibodies (Cell Signaling Technology, 5151S and 5257S), 1:5000 anti-rabbit HRP (Cell Signaling Technology, 7074), 1:5000 anti-mouse HRP (Cell Signaling Technology, 7076), 1:4000 anti-rabbit IRDye 800CW (LI-COR, 926–32211) or 1:4000 anti-mouse IRDye 680RD (LI-COR, 926–68070).

## TZM-bl infectivity assay

Media was recovered approximately 48 hr post-transfection and cell-free virus stocks were generated by filtering the media through 0.45 μM filters (Millipore). The TZM-bl indicator cell line was used to quantify the amount of infectious virus (*Derdeyn et al., 2000*; *Platt et al., 1998*; *Wei et al., 2002*). TZM-bl cells were seeded at 70% confluency in 24-well plates and infected by overnight incubation with virus stocks. 48 hr post infection, the cells were lysed and infectivity was measured by analyzing β-galactosidase expression using the Galacto-Star System following manufacturer's instructions (Applied Biosystems). β-galactosidase activity was quantified as relative light units per second using a PerkinElmner Luminometer.

## Immunoprecipitation assays

HEK293T cells in six-well plates were transfected with 800 ng of pKHNYN-1-FLAG, pKHNYN-2-FLAG, pGFP-FLAG as a control using 3 μL *Trans*IT-LT1 per 1 μg of DNA added. For the experiments in which lysates were treated with Ribonuclease A (RNase A), 500 ng of pHA-ZAP-L (*Kerns et al., 2008*) was also added. The cells were lysed on ice in lysis buffer (0.5% NP-40, 150 mM KCl, 10 mM HEPES pH 7.5, 3 mM MgCl$_2$) supplemented with complete EDTA-free Protease inhibitor cocktail

tablets (Sigma-Aldrich), 10 mM N-Ethylmaleimide (Sigma-Aldrich) and PhosSTOP tablets (Sigma-Aldrich). The lysates were sonicated and then centrifugated at 20,000 x g for 5 min at 4°C. 50 μl of the post-nuclear supernatants was saved as the input lysate and 450 μl were incubated with either 18 μg of anti-Flag antibody (Sigma-Aldrich, F7425) or 4.275 μg of anti-ZAP antibody (Abcam) for one hour at 4°C with rotation. Protein G Dynabeads (Invitrogen) were then added and incubated for 3 hr at 4°C with rotation. The lysates were then washed four times with wash buffer (0.05% NP-40, 150 mM KCl, 10 mM HEPES pH 7.5, 3 mM MgCl$_2$) before bound proteins were eluted with Laemmli buffer and boiled for 10 min. When indicated, RNase A (Sigma-Aldrich) was added to the post-nuclear supernatant and incubated for 30 min at 37°C. Protein expression was analyzed via western blot as described above.

## RNA purification and quantitative RT-PCR

Total RNA was isolated from transfected HeLa cells using a QIAGEN RNeasy kit accordingly with the manufacturer's instructions. Viral RNA was extracted from cell supernatants using a QIAGEN QIAamp Viral mini kit accordingly with the manufacturer's instructions. 500 ng of purified cellular RNA was reverse transcribed using random hexamer primers and a High-Capacity cDNA Reverse Transcription kit (Applied Biosystems). Quantitative PCR was performed using a QuantiStudio 5 System (Thermo Fisher). For genomic RNA and *nef* mRNA in the cell lysate, the HIV-1 RNA abundance was normalized to GAPDH levels using the GAPDH Taqman Assay (Applied Biosystems, Cat# Hs99999905_m1). For the genomic RNA in the media, absolute quantification was determined using a standard curve of the HIV-1 provirus DNA plasmid. The genomic RNA primers were GGCCAGG-GAATTTTCTTCAGA/TTGTCTCTTCCCCAAACCTGA (forward/reverse) and the probe was FAM-ACCAGAGCCAACAGCCCCACCAGA-TAMRA. The *nef* mRNA primers were GGCGGCGAC TGGAAGAAGC/GATTGGGAGGTGGGTTGCTTTG-3' (forward/reverse) (*Jablonski and Caputi, 2009*).

## Microscopy

Cells were seeded on 24-well plates on coverslips pre-treated with poly-lysine. HEK293T cells expressing a control guide RNA targeting the LacZ gene or a guide RNA targeting ZAP were transfected with 250 ng of pKHNYN-FLAG. 24 hr post-transfection, the cells were fixed with 4% paraformaldehyde for 15 min at room temperature, washed with PBS, and then washed with 10 mM glycine. The cells were then permeabilized for 15 min with 1% BSA and 0.1% Triton-X in PBS. Mouse anti-FLAG (1:500) and rabbit anti-ZAP (1:500) antibodies were diluted in PBS/0.01% Triton-X and the cells were stained for 1 hr at room temperature. The cells were then washed three times in PBS/ 0.01% Triton-X and incubated with Alexa Fluor 594 anti-mouse and Alexa Fluor 488 anti-rabbit antibodies (Molecular Probes, 1:500 in PBS/0.01% Triton-X) for 45 min in the dark. Finally, the coverslips were washed three times with PBS/0.01% Triton-X and then mounted on slides with Prolong Diamond Antifade Mountant with DAPI (Invitrogen). Imaging was performed on a Nikon Eclipse Ti Inverted Microscope, equipped with a Yokogawa CSU/X1-spinning disk unit, under 60-100x objectives and laser wavelengths of 405 nm, 488 nm and 561 nm. Image processing and co-localization analysis was performed with NIS Elements Viewer and Image J (Fiji) software.

## Analysis of CpG frequency in HIV-1, MLV and Sindbis virus

The 'analyze base composition' tool in MacVector was used to calculate the CpG frequencies for the HIV-1$_{NL4-3}$ (NCBI accession number M19921) genomic RNA, MLV genomic RNA (J02255) and Sindbis virus (NC_001547). The CpG frequencies were calculated using the following formula: number of CpG occurrences / (frequency of C * frequency of G) where frequency of the base is the number of occurrences of the base/total number of bases in sequence.

## Sindbis virus replication assays

Sindbis virus (SINV), a kind gift from Penny Powell (University of East Anglia), was expanded and titrated in BHK-21 cells (*Mazzon et al., 2018*). Control, ZAP, TRIM25 or KHNYN HeLa cells were plated at 100,000 cells/well in 12-well plates. The following day, the cells were infected with Sindbis virus at a multiplicity of infection of 0.005 pfu/cell. After 90 mins, the infectious media was removed, the cells were washed once with PBS and then incubated with 1 ml of media. The media from the

infected cells was harvested at 8, 16, 24 and 32 hr post-infection. 100 µl of serial diluted media from the cells (from $10^{-1}$ to $10^{-8}$) were added onto BHK-21 cells in 96 well plates (8,000 cells/well plated the previous day). After 90 min, the media was removed and replaced with fresh media. An MTT assay was carried out on each plate 24 hr later. Briefly, 20 µl of 50 mg/ml Thiazolyl Blue Tetrazolium Bromide in PBS were added onto the cell media for 2 hr at 37°C, after which the supernatant is removed and replaced with 40 µl of a 1:1 solution of isopropanol and DMSO. 20 min later, 35 µl of the supernatant are transferred onto a 96 well plate and signal read at 570 nm. Values from this assay were used to determine the TCID50 and pfu/ml.

## Statistical analysis

Statistical significance was determined using unpaired two-tailed t tests calculated using Microsoft excel software. Data are represented as mean ± SD. Significance was ascribed to p values $p < 0.05$.

## Acknowledgements

We thank other members of the Neil and Swanson laboratories for helpful discussions. The following reagents were obtained through the NIH AIDS Research and Reference Reagent Program, Division of AIDS, NIAID, NIH: TZM-bl from Dr. John C Kappes, Dr. Xiaoyun Wu and Tranzyme Inc; HIV-1 p24 Hybridoma (183-H12-5C) from Dr. Bruce Chesebro. The Antiserum to HIV-1 gp120 #20 (ARP421) was obtained from the NIBSC Centre for AIDS Reagents. Dr. Harmit Malik kindly provided the ZAP-L expression vector and Dr. Jonathan Stoye kindly provided the anti-MLV p30 antibody. These studies were funded by MRC Discovery Award MC/PC/15068 and a Wellcome Trust Senior Research Fellowship (WT098049AIA) to SJDN, Medical Research Council grant MR/M019756/1 to CMS and Medical Research Council grant MR/S000844/1 to SJDN and CMS. MF is supported by the UK Medical Research Council (MR/R50225X/1) and is a King's College London member of the MRC Doctoral Training Partnership in Biomedical Sciences. MiM and MaM are supported by Medical Research Council funding to the MRC-UCL LMCB University Unit (MC_UU_12018/1). This work was also supported by the Department of Health via a National Institute for Health Research Comprehensive Biomedical Research Centre award to Guy's and St. Thomas' NHS Foundation Trust in partnership with King's College London and King's College Hospital NHS Foundation Trust.

## Additional information

### Funding

| Funder | Grant reference number | Author |
| --- | --- | --- |
| Medical Research Council | MC/PC/15068 | Stuart JD Neil |
| Wellcome | WT098049AIA | Stuart JD Neil |
| Medical Research Council | MR/M019756/1 | Chad M Swanson |
| Medical Research Council | MR/R50225X/1 | Mattia Ficarelli |
| Medical Research Council | MR/S000844/1 | Stuart JD Neil Chad M Swanson |
| Medical Research Council | MC_UU_12018/1 | Mark Marsh Michela Mazzon |

The funders had no role in study design, data collection and interpretation, or the decision to submit the work for publication.

### Author contributions

Mattia Ficarelli, Conceptualization, Formal analysis, Validation, Investigation, Writing—review and editing; Harry Wilson, Rui Pedro Galão, Formal analysis, Validation, Investigation, Writing—review and editing; Michela Mazzon, Irati Antzin-Anduetza, Formal analysis, Investigation, Writing—review and editing; Mark Marsh, Conceptualization, Supervision, Writing—review and editing; Stuart JD Neil, Conceptualization, Formal analysis, Supervision, Funding acquisition, Project administration, Writing—review and editing; Chad M Swanson, Conceptualization, Resources, Formal analysis,

Supervision, Funding acquisition, Writing—original draft, Project administration, Writing—review and editing

**Author ORCIDs**
Mattia Ficarelli (ID) https://orcid.org/0000-0002-9380-132X
Harry Wilson (ID) https://orcid.org/0000-0002-3185-1073
Rui Pedro Galão (ID) https://orcid.org/0000-0003-3368-5053
Michela Mazzon (ID) http://orcid.org/0000-0002-2462-9925
Stuart JD Neil (ID) https://orcid.org/0000-0003-3306-5831
Chad M Swanson (ID) https://orcid.org/0000-0002-6650-3634

**Decision letter and Author response**
Decision letter https://doi.org/10.7554/eLife.46767.021
Author response https://doi.org/10.7554/eLife.46767.022

## Additional files

**Supplementary files**
• Supplementary file 1. Key resources table.
DOI: https://doi.org/10.7554/eLife.46767.018

• Transparent reporting form
DOI: https://doi.org/10.7554/eLife.46767.019

**Data availability**
All data generated or analysed during this study are included in the manuscript and supporting files.

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
