## [Decision Letter]

Thank you for submitting your article "KHNYN is essential for ZAP-mediated restriction of HIV-1 containing clustered CpG dinucleotides" for consideration by *eLife*. Your article has been reviewed by Päivi Ojala as the Senior Editor, a Reviewing Editor, and two reviewers. The following individuals involved in review of your submission have agreed to reveal their identity: Michael Emerman (Reviewer #1); Eric Freed (Reviewer #2).

The reviewers have discussed the reviews with one another and the Reviewing Editor has drafted this decision to help you prepare a revised submission.

Summary:

In this study, Ficarelli and colleagues identify KHNYN, a protein with no previously defined function, as a co-factor in ZAP-mediated degradation of viral RNAs containing CpG dinucleotides. While other co-factors for ZAP have previously been described, the factor encoding a putative RNA-degrading activity had not been found. The authors identify KHNYN as a ZAP-interacting protein in a yeast two-hybrid screen, confirming the results of previous screens for ZAP interactors performed by other groups. Overexpression of KHNYN resulted in the degradation of an HIV-1 variant engineered to contain 36 introduced CpG dinucleotides but had only a small effect of WT HIV-1, which, like many viral RNA genomes, is underrepresented for CpGs. Reductions in RNA levels were accompanied by reductions in viral protein production. The effect of KHNYN overexpression was found to be dependent on ZAP and also on expression of TRIM25, confirming earlier findings that TRIM25 is involved in ZAP-mediated antiviral activity. The authors show that knock-out of KHNYN prevented ZAP-mediated degradation of the CpG-containing HIV-1 RNA.

In summary, this is a very nice study with straightforward assays showing that KHNYN is essential for this function of ZAP. The data are clear-cut and of high quality, and add important information to the mechanism of ZAP-mediated antiviral activity. The narrow focus on a high CG-containing HIV-1, an unnatural target of ZAP, limits the conclusions and slightly reduces the overall high impact of the study. This work will, nonetheless, be of great interest to a broad audience well beyond the HIV field.

Essential revisions:

1) The authors convincingly show that KHNYN is required and necessary for ZAP to inhibit a high CG-containing HIV. While this makes a nice model, it is not a natural target of ZAP. The authors could broaden their claims by using other viruses to show whether or not KHNYN is needed for ZAP-mediated inhibition of other targets; for example, MLV would work, and/or alphaviruses. The existing knockout cells should be sufficient for these assays.

2) The authors confine their studies to HeLa and 293 cells. It is possible that the factor needed to degrade high CG RNA via ZAP is different in different cells. Specifically, the authors should look at OhAinle et al., 2018, especially the data and text describing ZAP and N4BP1 in THP-1 cells. The authors are welcome to these reagents if they want them.

N4BP1 is the other NYN protein described in the Marco and Marin, (2009). While additional experiments are not expected here, the authors should discuss the possibility that the effector protein needed for ZAP might be different in different cells and could confer different specificity on the CG-containing RNAs that are degraded (also note that N4BP1 is on the list of interactors in the Youn et al., 2018).

3) In Figure 1 please show the entire dataset of high confidence interactors unless KHNYN is the only one. What is the overlap between this list and the ZAP-interactors identified by BioID (Youn et al., 2018)?

4) The authors need to more thoroughly examine effects of KHNYN on RNA levels. While most of the assays used are measuring protein or infectivity, the essential data for the proposed mechanism is examining RNA levels. Figure 8A is the only one that measures this, but is not quite satisfying. Importantly, they should show levels of different spliced forms of HIV-1 RNA. Presumably, the Nef spliced message should not be decreased and would be a good control. The authors should also present the data as actual levels of HIV-1 RNA normalized to a cellular message, rather than only relative RNA abundance of HIV-1.

5) The authors state that they do not have a good anti-KHNYN antibody to detect endogenous protein levels in their KO cells, but the KOs should be analyzed by providing the genomic analysis in the supplement to determine the percentage knockouts in the pool and to show the nature of the lesions in the clones.

---

## [Author Response]

Summary:[…] In summary, this is a very nice study with straightforward assays showing that KHNYN is essential for this function of ZAP. The data are clear-cut and of high quality, and add important information to the mechanism of ZAP-mediated antiviral activity. The narrow focus on an high CG-containing HIV-1, an unnatural target of ZAP, limits the conclusions and slightly reduces the overall high impact of the study. This work will, nonetheless, be of great interest to a broad audience well beyond the HIV field.Essential revisions:1) The authors convincingly show that KHNYN is required and necessary for ZAP to inhibit a high CG-containing HIV. While this makes a nice model, it is not a natural target of ZAP. The authors could broaden their claims by using other viruses to show whether or not KHNYN is needed for ZAP-mediated inhibition of other targets; for example, MLV would work, and/or alphaviruses. The existing knockout cells should be sufficient for these assays.

We have analyzed the effect of knocking out KHNYN on both MLV and Sindbis virus using our ZAP and KHNYN CRISPR cells (new Figure 7D-E and Figure 7—figure supplement 3). ZAP has previously been shown to target MLV RNA for degradation. Depletion of KHNYN increases MLV Gag expression and virion production to similar levels as knocking out ZAP. This phenotype is particularly pronounced after treatment of the cells with type 1 interferon. Interestingly, KHNYN does not restrict Sindbis virus. Depletion of ZAP or TRIM25 increased viral replication as expected, but, in contrast, KHNYN depletion had no effect. Since the major role of ZAP in restricting Sindbis virus has been shown to be inhibition of viral RNA translation, this may indicate that ZAP uses different cofactors depending on whether it inhibits translation or targets the RNA for degradation.

2) The authors confine their studies to HeLa and 293 cells. It is possible that the factor needed to degrade high CG RNA via ZAP is different in different cells. Specifically, the authors should look at OhAinle et al., 2018, especially the data and text describing ZAP and N4BP1 in THP-1 cells. The authors are welcome to these reagents if they want them.N4BP1 is the other NYN protein described in the Marco and Marin, (2009). While additional experiments are not expected here, the authors should discuss the possibility that the effector protein needed for ZAP might be different in different cells and could confer different specificity on the CG-containing RNAs that are degraded (also note that N4BP1 is on the list of interactors in the Youn et al., 2018).

The N4BP1 data in OhAinle et al., is very interesting and we agree that this is an important concept. In the Discussion section, we have highlighted the N4BP1 results in the context of a paragraph on how ZAP could use different cofactors in different cell types. Of note, N4BP1 is not a ZAP-interacting protein in Youn et al., 2018, though it did interact with TRIM25. How KHNYN and N4BP1 mediate ZAP antiviral activity in different cell types in the absence and presence of type I interferon will be an important area of future investigation.

3) In Figure 1 please show the entire dataset of high confidence interactors unless KHNYN is the only one. What is the overlap between this list and the ZAP-interactors identified by BioID (Youn et al., 2018)?

For the yeast-two-hybrid screen to identify ZAP-interacting factors, KHNYN was the only scored hit for ZAP-L and ZAP-S. We have modified the text in the Results section relating to Figure 1 to describe this. It is unclear why other ZAP-interacting factors previously identified by BioID, affinity purification-mass spectrometry or co-immunoprecipitation were not identified in the yeast-two-hybrid screen, but the LexA system was specifically used because of its stringency.

4) The authors need to more thoroughly examine effects of KHNYN on RNA levels. While most of the assays used are measuring protein or infectivity, the essential data for the proposed mechanism is examining RNA levels. Figure 8A is the only one that measures this, but is not quite satisfying. Importantly, they should show levels of different spliced forms of HIV-1 RNA. Presumably, the Nef spliced message should not be decreased and would be a good control. The authors should also present the data as actual levels of HIV-1 RNA normalized to a cellular message, rather than only relative RNA abundance of HIV-1.

We agree that this is an important control and have added a panel to Figure 8 analyzing *nef* mRNA abundance (new Figure 8B). As expected, it does not differ between wild type HIV-1 and HIV-1 with clustered CpG dinucleotides. *nef* mRNA levels are also not affected by ZAP or KHNYN depletion.

The qRT-PCR results in the cell lysates (Figure 4B, Figure 8A and Figure 8B) is the HIV-1 genomic RNA or *nef* mRNA normalized to GAPDH. These values have then been used to calculate the relative abundance. We have modified the Materials and methods section to make this clear.

5) The authors state that they do not have a good anti-KHNYN antibody to detect endogenous protein levels in their KO cells, but the KOs should be analyzed by providing the genomic analysis in the supplement to determine the percentage knockouts in the pool and to show the nature of the lesions in the clones.

We have sequenced the lesions in the KHNYN CRISPR cells and identified the mutated regions, which are presented in Figure 7—figure supplement 2.